

# Scaling Artificial Heat Islands to Enhance Precipitation in Arid Regions

Oliver Branch, Lisa Jach, Thomas Schwitalla, Kirsten Warrach-Sagi, Volker Wulfmeyer

Institute of Physics and Meteorology, University of Hohenheim, Stuttgart, 70599, Germany

5 *Correspondence to*: Oliver Branch (oliver_branch@uni-hohenheim.de)



**Abstract.**

Potential for regional climate engineering is gaining interest as a means of solving regional environmental problems like water scarcity and high temperatures. In the hyper-arid United Arab Emirates (UAE), water scarcity is reaching crisis point due to high consumption and over-extraction, and is exacerbated further with climate change. To counteract this problem, the UAE has conducted cloud seeding operations and intensive desalination for many years, but is now considering other means of increasing water resources. Very large 'Artificial Black Surfaces' (ABS), made of black mesh/black painted/solar PV panels have been proposed as a means of enhancing convective precipitation, via surface heating and amplification of vertical motion. Under the influence of the daily UAE sea breeze, this can lead to convection initiation under the right conditions. Currently it is not known how strong this rainfall enhancement would be, nor what scale of black surface would need to be employed. This study simulates the impacts at different ABS scales using the WRF-NoahMP model chain and investigates impacts on precipitation quantities and underlying convective processes. Simulations of five square ABS of 10, 20, 30, 40, and 50 km sizes were made on four one-day cases over a 24-hour period. These were compared with a Control model run, with no land use change, to quantify impacts. The ABS themselves were simulated by altering land cover static data, and prescribing a unique set of land surface parameters like albedo and roughness length.

On all four days, rainfall is enhanced by low-albedo surfaces of 20 km or larger, primarily through a reduction of convection inhibition and production of convergence lines and buoyant updrafts. The 10 km square ABS had very little impact. From 20 km upwards there is a strong scale-dependency, with ABS size influencing the strength of convective processes and volume of rainfall. In terms of rainfall increases, the 20 km produces a mean rainfall increase, over the Control simulation, of 571,616 $m^3\,day^{-1}$, 30 km (~1 mil. $m^3\,day^{-1}$), 40 km (~1.5 mil. $m^3\,day^{-1}$), and 50 km (~ 2.3 mil. $m^3\,day^{-1}$). If we assume that such rainfall events happen only in 10 days in a year, this would equate to respective annual water supplies for >31,000, >50,000, >79,000, and >125,000 extra people $yr^{-1}$, at UAE per capita consumption rates. Thus, artificial heat islands made from black panels or solar PV offer a means of enhancing rainfall in arid regions like the UAE, and should be made a high priority for further research.



## 1  Background

Regional crises like high temperatures, drought, wildfires, flooding and water scarcity are become more severe, and the
potential for regional climate engineering is gaining interest. Examples are marine cloud seeding to reduce coral bleaching
(Latham et al., 2014; Latham et al., 2013; Tollefson, 2021) and albedo management through landscape planning, and breeding
of higher-albedo crops (Doughty et al., 2011; Kala et al., 2022; Ridgwell et al., 2009). Albedo change can trigger regional
scale impacts (Quaas et al., 2016; Seneviratne et al., 2018) such as reduction of temperatures (Kala & Hirsch, 2020), but at the
same time could also contribute toward reduced global forcing (Carrer et al., 2018; Sieber et al., 2022).

Modified albedo could also influence precipitation in arid regions. In the Middle Eastern Gulf region, water scarcity under a
changing climate is reaching crisis point, and diverse methods for obtaining water are now considered, including cloud seeding
(e.g. Al Hosari et al., 2021) and importation of water and even icebergs (Condron, 2023; Abdulrahman, 2020). Other avenues
are also possible. Several studies show that desert xerophyte plantations can enhance rainfall via canopy heating (Becker et
al., 2013; Branch et al., 2014; Branch & Wulfmeyer, 2019; Wulfmeyer et al., 2014). Branch et al., 2014, measured albedos of
0.17 and 0.12 for jatropha and jojoba plants, and the surrounding desert ~0.3, leading to temperatures up to 4°C higher than
the surrounding desert (see also Saaroni et al., 2004). This heating led to greater simulated cloud development and convection
initiation (CI) (Branch & Wulfmeyer, 2019). A related method was proposed by Black & Tarmy (1963), who claimed a similar
effect from asphalt patches, and defined relations between vertical updraft penetration and patch length. Asphalt is not likely
to be desirable in rural areas, but presumably solid or mesh black panels would also work and be more easily de-installed.
Panels could be darkened with black paint of albedo ~5% could be used to coat the panels, or even specialist coatings with <
1% albedo (Theocharous et al., 2014). Other surfaces which may modify weather are solar PV panels (Li et al., 2018; Lu et
al., 2021; Mostamandi et al., 2022). These are typically of the mono- or poly-crystalline type, with very low albedos of between
~0.04 (Nemet, 2009) and ~0.1 (Zachariou & Protogeropoulos, 2010). The latter study found that albedo values can increase
with high zenith angles due to glare effects though. With PV, one must also account for radiation converted to electrical power,
which for a given radiation flux would leave less energy for sensible heating. The net total can be expressed as an *effective*
albedo, here in Eq. (1):

$$A_{eff} = \delta + \varepsilon, \tag{1}$$

where $\delta$ and $\varepsilon$ are reflectivity and conversion efficiency, respectively (Taha, 2013). PV efficiency can theoretically reach
~46% in laboratory conditions (Allouhi et al., 2022), but in reality, is usually closer to 10-20%, with most radiation transformed
into heating (Taha, 2013). This may be useful for rainfall enhancement, but can reduce cell efficiency and longevity (Dwivedi
et al., 2020). Assuming the lowest albedo of 0.04, and panel efficiency between 0.1 and 0.15, this would yield effective albedos
of 0.14 and 0.19, i.e., similar to jojoba and jatropha. For simplicity in this study we will use an umbrella term 'Artificial Black
Surfaces (ABS)' for these systems, whether they are made of black panels, PV or any composite of such surfaces.



When comparing the potential for ABS and plantations to modify weather it is important to consider differences in surface characteristics. ABS is likely to have lower albedo than plants, zero leaf transpiration, and different roughness characteristics (depending on panel material and geometry), and dissimilar under-panel shading, and longwave and ground heat fluxes. Taken together, these may produce unique land-atmosphere interactions and these must be investigated, potentially with model simulations. An ensemble of simulations could provide insights into impacts on convective processes, and the influence of

ABS scale.

In terms of convective processes, there may be some similarities with the urban heat island (UHI) effect and there is a range of UHI process studies using analytical methods (e.g., Fan et al., 2016; Han & Baik, 2008; Hidalgo et al., 2010; Lee & Olfe, 1974; Lee et al., 2012), laboratory experiments (e.g., Cenedese & Monti, 2003; Jie Lu et al., 1997), and modelling/observational studies (Stewart et al., 2021; Zhang et al., 2004). Some studies include the interactions between UHIs

and sea breezes (e.g., Cenedese & Monti, 2003; Freitas et al., 2007; Zhang & Wang, 2021), and found interactions between them, particularly for convection. Such interactions will become relevant for this study. However, there are some dissimilarities between urban areas and ABS too. Urban heating often develops later in the day due to urban heat storage (Soltani & Sharifi 2017), whereas plantation heating occurs and dissipates earlier (Branch et al. 2014). The heterogeneous horizontal and vertical structure of cities would also differ from flat, homogeneous ABS panels at heights of ~1-2 m. Additionally, cities often contain

green areas or water bodies, which may produce more spatially-heterogeneous atmospheric exchanges (Chrysoulakis et al., 2018; Offerle et al., 2005; Voogt & Oke, 2003). Urban areas are often simulated with complex urban models (e.g., see Salamanca et al., 2018; Vogel & Afshari, 2020), but given the more homogeneous structure of ABS, a more simplified approach may be taken.

We employ a mesoscale model case-study approach to investigate scale-dependent ABS impacts on precipitation and

convective processes. The structure of the work is as follows: the simulation strategy, weather conditions, and case selection are presented in Materials and Methods. In the Results and Discussion section, we present impacts on precipitation, convective processes, and feedbacks. Finally, in the Summary and Outlook section we put results in context and discuss wider implications.

## 2   Material and Methods

### 2.1 Model Configuration

Simulations were carried out with the Weather Research and Forecasting (WRF) model (V4.2.1, Powers et al., 2017). WRF has been used in the region for numerous model evaluation and process (Branch et al., 2021; Fonseca et al., 2020; Valappil et al., 2020; Wehbe et al., 2019; Nelli et al., 2020; Schwitalla et al., 2020), and rainfall modification studies (Mostamandi et al., 2022; Wulfmeyer et al., 2014; Branch & Wulfmeyer, 2019). Here, we use the same domain, resolution and configuration to

Branch et al., 2021, who evaluated the skill of WRF (V3.8.1) in reproducing 2-m temperature and humidity when compared to 48 surface weather stations. The only change is that here an updated version of WRF was used to take advantage of



improvements to the MYNN planetary boundary layer (PBL), surface layer, dynamics and land surface schemes, as well as vapor mixing ratio (Q-2m) diagnostics. Table 1 shows the selected physics schemes.

**Table 1: Selected physics schemes used on WRF for the simulations.**

| Physics type | Scheme/Option | Reference |
|---|---|---|
| Land surface | Noah-MP | Niu, 2011 |
| Atmospheric surface layer | MYNN | Nakanishi & Niino, 2006 |
| Planetary boundary layer | MYNN 2.5 level TKE | Nakanishi & Niino, 2006 |
| SW radiation | RRTMG | Mlawer et al., 1997 |
| LW radiation | RRTMG | Iacono et al., 2008 |
| Microphysics | Morrison 2-moment | Morrison & Gettelman, 2008 |

## 2.2 Domain and Grid Configuration

Figure 1, panel (a) shows the model domain (dotted line) covering the whole of the UAE on the northeast coast of the Arabian Peninsula, plotted on terrain height. The grid increment is based on a latitude-longitude geographic coordinate system, spanning 43.275° to 65.725°E and 14.775 to 32.225°N with a spacing of 0.025° ($\Delta x$ 2.779 km). This grid spacing provides a good representation of convective processes and landscape structure, whilst optimizing computational demands for running numerous model realizations. The grid has dimensions of $899(x) \times 699(y) \times 100(z)$. The model levels were configured such that typical boundary layer depths contained at least 25 levels, and the center of the lowest model level is ~15 m above ground level (AGL). A large domain was used to capture the prevailing northerly winds and sea breeze. Large-scale forcing data was retrieved from the European Centre for Medium Range Weather Forecasts (ECMWF) Integrated Forecasting System (IFS), in the form of 6-hourly operational-analysis model-level data on the 41r1 cycle ($\Delta x$ 0.125°) grid. Soil moisture and temperatures are provided by the model, which then assimilates satellite data (Albergel et al., 2012) into the land surface model HTESSEL (Balsamo et al., 2009). Additionally, OSTIA sea surface temperature (SST) data ($\Delta x$ 1/20°, Donlon et al., 2012) were also ingested every 12 hours (00:00 and 12:00 UTC), which is particularly important for simulating sea breezes.





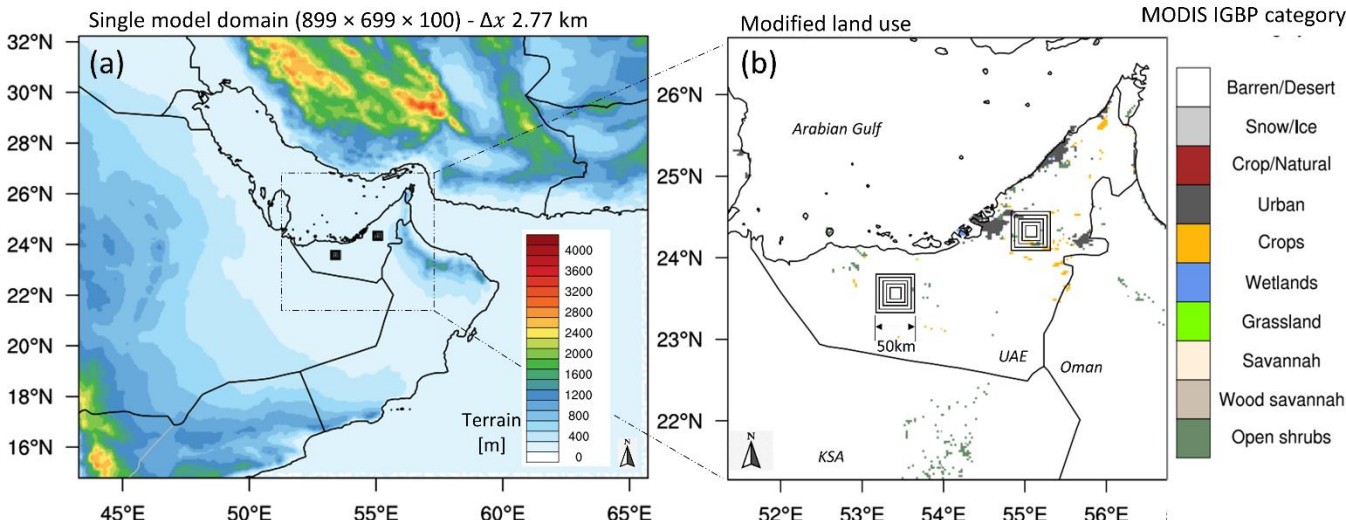

**Figure 1: WRF model domain for all simulations, and the different scenarios used. Panel (a) shows the terrain over the northern Arabian Peninsula, and the inset model domain (2498 × 1942 km) centered on the United Arab Emirates (UAE) [dashed rectangle]. Panel (b) is the full extent of the model domain (Δ𝑥 2779 m) showing MODIS IGBP land cover data. Also shown here are the five artificial-surface scenarios used in the form of pairs of different-sized squares (50×50, 40×40, 30×30, 20×20, and 10×10 km).**

### 2.3 Modelling Strategy and ABS Simulation

The ABS simulations are based on an idealized panel of low-roughness length and a low-albedo (0.05), which aligns more or less with acrylic black paint (see Table 2 for all parameters).

We simulated five separate ABS scenarios with sizes: 10×10, 20×20, 30×30, 40×40, and 50×50 km, with two ABS in each scenario to assess impacts in two locations (*east* and *west*). The ABD zones were input into WRF as modifications the land cover static data (Figure 1, panel b). The ABS centers were collocated to maximize comparability; spaced some distance apart (>60 km) to minimize the influence between them; and placed >50 km from the coast, to investigate initial impacts away from the sharp coastal temperature/humidity gradients. All scenarios were compared with a *Control* simulation, i.e., no land use change, to quantify ABS impacts and also to characterize the weather conditions. Each of the four case studies was run for a 24-hour period (00:00 to 00:00 UTC).

**Table 2: Land surface parameters chosen to represent a flat black surface (10 cm thick) and 50 cm off the ground.**

| Land surface Parameter | Prescribed value for ABS |
|---|---|
| Momentum roughness length [m] | 0.05 m |
| Top of canopy [m] | 0.5 m |
| Base of canopy [m] | 0.4 m |
| Vegetation status [-] | None |
| Stem Area Index [-] | 0 |
| Leaf Area Index [-] | 0 |
| Background Albedo [-] | 0.05 |
| Min. Stomatal Resistance [s m$^{-1}$] | 9999 |



## 2.4 Case Selection


The UAE summer (June, July and August (JJA)) climate is characterized by high temperatures and low precipitation, influenced by subtropical subsidence and inversions. Occasionally though, southerly monsoon flows can transport moisture to the eastern UAE (Böer, 1997; Schwitalla et al., 2020) causing sporadic convection (Branch et al., 2020). The prevailing JJA winds are northerly, so the Arabian Gulf has a strong influence on the weather. The coastal Arabian Gulf is relatively shallow

(maximum depth ~90 m) and heats rapidly, with SSTs often >30°C (Al Azhar et al., 2016). Warm sea breezes reach the coast around midday (local time (LT), Eager et al., 2008), and reach the ABS areas typically around 14:00 LT.

Our aim was to observe strong impacts, so we selected a season where impacts are most likely days with unstable convective conditions during a suitable season. Summertime (JJA) was selected because strong impacts were observed during this season (e.g. Branch and Wulfmeyer, 2019) and we elected to do the simulations in JJA 2015, to coincide with the evaluation

simulations of Branch et al. (2021), who also found this season was representative in terms of the long-term climate.

We illustrate weather conditions for the four selected cases in Figures 2 and 3 (from Control) to highlight the likelihood of impacts. Figure 2 shows Skew-T diagrams at two times - 06:00 and 10:00 UTC (LT = UTC+4). Typical CI onset times in the UAE is ~07:30-09:30 UTC (Branch et al., 2020). The first three rows (18,24,25 July) are soundings over the *west* ABS, and the last row (27 July), the *east* ABS (the reason relates to observed impacts - explained later in Results and Discussion). Figure

3 shows a horizontal perspective of convective available potential energy (CAPE), convective inhibition (CIN), and 10-m winds (10:00 UTC). The sea breeze is evident in both figures, arriving at the ABS at ~09:00–10:00 UTC. In Figure 2, surface winds at 06:00 UTC are strong and north/north-easterly on 18, 24, 25 July (panels a, c, and e). By 10:00 UTC, these winds swing northwards, and also at greater height (850 hPa) as the breeze strengthens. On 27 July winds are south/westerly over the east ABS, and remain so at 10:00.






**Figure 2: Skew-T plots showing thermodynamic profiles for four cases to show pre-convective conditions. Left-hand blue curves represent vapor mixing ratio (reference to isohumes) and dewpoint (isotherms). Isohumes are green dotted lines rising left to right (values marked in blue, g kg⁻¹). Red curves show temperature (red text on x axis, °C). CAPE is the area bounded by the red dotted line and the vertical temperature profile. Horizontal isobars are constant pressure (hPa). Dry adiabats are brown isolines rising right to left. Moist adiabats are solid green lines rising right to left. Soundings are from a single grid cell in each WRF Control run, corresponding to the center of the east (27 July) or west ABS (18, 24, 25 July). The choice of location is discussed later. On the right are wind barbs. The east sounding is located at 24.341°N, 55.061°E and the west sounding at 23.584°N, 53.388°E.**




In Figure 3, the northerly sea breeze is evident, particularly to the west. In the east, conditions vary more with influence from
the Gulf of Oman. Sometimes wind confluence occurs (e.g., 27 July, Figure 3, panel d), or one side dominates the other, or
more southerly winds prevail.

In Figure 2, the 06:00 show stable temperature layers at 900-850 hPa, which can inhibit convection. On 18 July and 27 July
these are more like inversions, whereas on 24 and 25 July it is more unstable. By 10:00 the stable layers/inversions disappear
due to increased mixing. In all cases, the 10:00 temperature profiles exceed a conditionally-unstable state (~dry adiabatic lapse
rate). In Figure 3, the sea breeze brings high CAPE (>~700 J kg$^{-1}$, lower panels), and lower CIN (<~50 J kg$^{-1}$) (lower panels).
On 18,24,25 July, this influx is more apparent to the west. On 27 July, it occurs all along the coast. In Figure 2 (06:00), surface
temperatures are already high ($\geq$ 40°C) and increase further by 10:00. Moisture is more variable, with some 06:00 stratification
particularly on 18 July which has a dry layer above 900 hPa. None of the days is overly humid, and values reach only 9-10 g
kg$^{-1}$. As expected, moisture is also well-mixed by 10:00, with the exception of 18 July which remains dry above 900 hPa. We
may expect this case to have only weak impacts.

In summary, all cases are moderately unstable in all cases and have a wave of reduced CIN passing over the ABS. CIN may
be a defining constraint, for even when CAPE is only moderate (e.g., 27 July), a low CIN may still permit CI.





**Figure 3: Control thermodynamic conditions at 10:00 am (UTC) during the four case studies, with red boxes to show the relative position of the ABS. The top row shows convective available potential energy (CAPE, J kg$^{-1}$), 10 m wind vectors, and a box showing the vector reference length for 3 m s$^{-1}$ (panel (a)). The bottom row shows convective inhibition (CIN, J kg$^{-1}$). These conditions were used to investigate the daily sea breeze timing and select our case studies.**



## 3 Results and Discussion

### 3.1 Precipitation

We first examine precipitation impacts (in mm d$^{-1}$, or in m$^3$ for volumes). During analysis, we imposed two constraints. Firstly, we apply a commonly-used filter of 1 mm day$^{-1}$ to remove model drizzle (Frei et al., 2003). Secondly, we disregarded all rainfall beyond a certain distance from the ABS, to ensure that differences are caused by physical processes and not numerical errors as discussed by Ancell et al. (2018) which can travel at speeds >3600 km h$^{-1}$ and bias the results of impact studies. We selected a radius of 90 km from the ABS center as a reasonable distance with respect to typical advection velocity, and excluded precipitation from outside these circles. This radius is somewhat arbitrary and it is possible that some precipitation impacts extend beyond the perimeter, but it is equivalent for all scenarios.

Precipitation impacts are quantified in Figures 4 and 5. Figure 4 shows 24-hour accumulated precipitation for four scenarios (Control, 10, 30, and 50 km). There is an increase in rainfall coverage and maximum rain rates (blue-to-red) in all four cases when moving from Control > 10 > 30 > 50 km (strongest impacts are marked with red circles). In the first three cases (18-25 July) significant impacts occur around the west ABS, but on 27 July, impacts occur near the east ABS – this is the reason for our choice of sounding locations in Figure 2. The impact is weakest on 18 July, which is not surprising given the inversion and dry layer. We also discuss later how this relates to the convection onset time. What is clear is that much of the surplus rain falls outside of the square ABS perimeters, mainly on the leeside with respect to wind direction.




**Figure 4: Total precipitation (mm) over 24 hours for each case study (00:00 – 23:30 UTC). The four rows show the Control, 10 km, 30 km, and 50 km scenarios, respectively – 20 km and 40 km not shown here). The solid black squares indicate the ABS zones. The dotted squares in Control show hypothetical surfaces, just for visual reference. The dotted black circles are 150 km diameter circles within which the rainfall volume was calculated (m³). This limitation was imposed to discard propagated numerical differences related to initial conditions. Red circles show areas of high interest, i.e. where the most visible impacts occurred.**

Figure 5 shows accumulated water volume (sum of east and west circles). The onset time for precipitation for 24, 25, and 27 July is ~09:00, and for 18 July ~09:30, which aligns with the sea breeze arrival times. There is some precipitation in Control, but the amounts are enhanced by all ABS from the 20 km scale upwards. The 10 km ABS impact is almost negligible, except for 24 July which enhances the amount by ~0.4 mil. m³ (panel e).  On 25 July, the 10 km amount is even lower than Control.



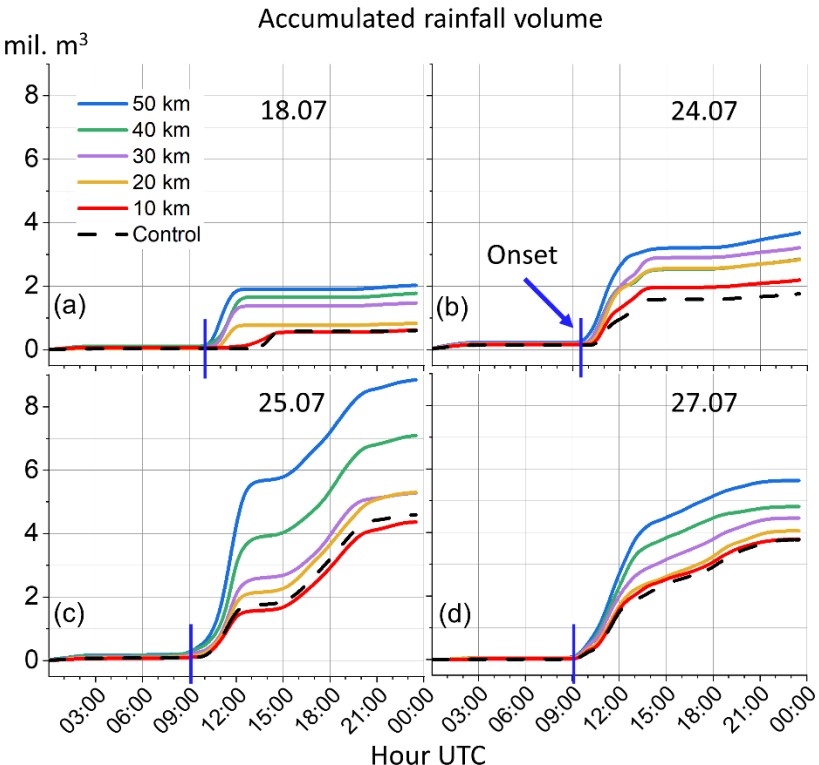

**Figure 5: Spatially-integrated precipitation volume within the 180 km diameter circles shown in Figure 4. Panels (a-d) show the six scenarios for each of the four cases.**

We can also express the water increase in domestic consumption units. We first show the separate cases and scenarios in Figure 6. Panel (a) shows the total daily water volumes. Panel (b) shows the difference between the ABS and Control expressed in *volume* (left axis, mil. m³) and UAE per capita *supply* based on 500-liter capita⁻¹ day⁻¹ (right axis, people yr⁻¹), amongst the highest in the world (Albannay et al., 2021; Yagoub et al., 2019).  In panel (a), there are steady increases with scale for the most part, with an approximate doubling of the Control rainfall in the 50 km scenario. We can see the case differences, but the

impacts appear to increase with ABS scale, although this relationship does not always hold (e.g., on 24 July, the 40 km ABS has less impact than the 30 km ABS). Again, we can see that the 10 km scale has no consistent impact, with even a slight decrease on the 25 July. There is a small surplus on 24 July though. From panel (b), we ranked the cases in terms of volume increase between the 50 km scenario and Control. By doing so, we may later investigate if there are process-based reasons for the differences. The cases were ranked as: 1) 25 July [4.25 mil m³], 2)  24 July [1.92 mil m³], 3) 27 July [1.87 mil m³], and 4)

18 July (1.42 mil m³). Clearly the 25 July produced by far the largest increase in water volume, followed by 24 and 27 July (which are almost the same), and finally by 18 July which had a smaller increase.



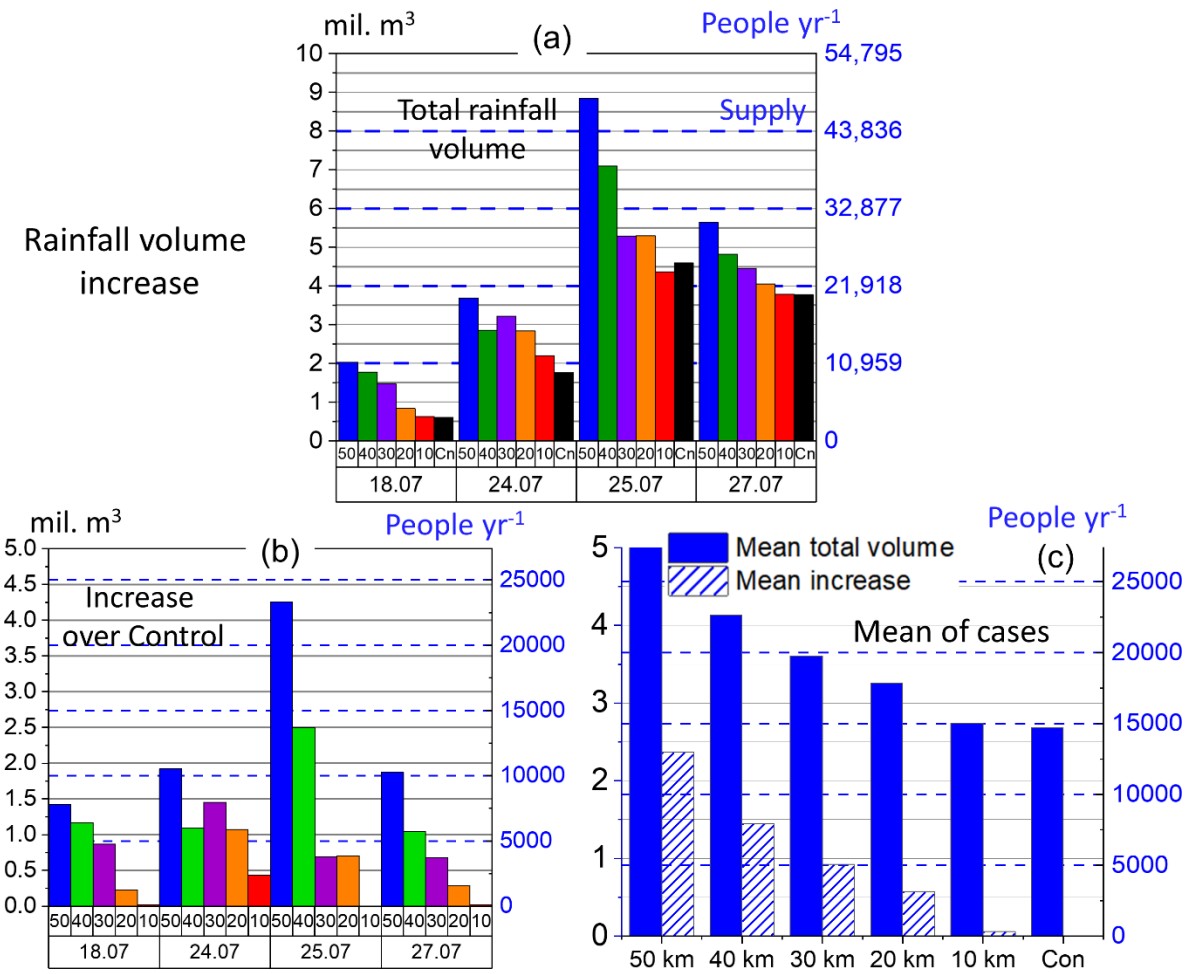

**Figure 6: Total daily water volumes and increases. Panel (a) shows the total water volume that fell within the 180 km diameter**
**circles. These is expressed as a volumetric amount (left axis, mil. m³), and as an equivalent annual supply per capita in the UAE**
**(right axis, people yr⁻¹), based on 500 L day⁻¹ capita⁻¹ consumption. Panel (b) shows the difference in volume between the case and**
**the Control. Panel (c) shows the averages for the cases. The solid bars show the mean volume over all cases, and the striped bars are**
**the mean increase over Control.**

As well as quantifying the increases for individual cases, we also computed the mean increases for all cases to infer the 'typical'

impact of a rainfall event (Figure 6). Panel (c) shows the mean total water volume (solid bars) and then the mean increases

over Control (striped bars). The mean values indicate a clearer relationship between scale and surpluses. It is not linear, but

more exponential in appearance – which is reasonable, given the areal size increments. We can also see that 20 km appears to

be a lower boundary for impacts, producing a mean increase of 571,616 m³ per rainfall event (i.e., 3132 people yr⁻¹). At this

scale, if such an event occurred just 10 times in one summer, this would produce >5 mil. m³ extra water, and supply >31,000

extra people yr⁻¹. A 30 km ABS could produce >10 mil. m³ extra water and supply >50,000 extra people yr⁻¹. A 50 km ABS





could produce >20 mil. m³ extra water and supply > 125,000 extra people yr⁻¹. These are significant amounts, which may be exceeded if events occur more often, perhaps in spring and autumn.

Of course, not all rainwater is likely to be recoverable for domestic consumption, so these amounts are somewhat illustrative. On the other hand, designs for water recovery have been considered (European Patent-EP3909659). In reality, most of the
rainfall that reaches the ground will either evaporate, form runoff, be taken-up/intercepted by plants, or percolate into the soil. Water that percolates deeper into the soil can help to recharge groundwater or even aquifers, which are already depleted due to groundwater extraction or degraded by sea-water intrusion (Sefelnasr et al., 2022; Sherif et al., 2011, 2023; Sherif & Kacimov, 2007). Desalinated groundwater is routinely used for agriculture, industrial or domestic use in the UAE, so an increase in groundwater would be beneficial. Additionally, increased rainfall could offset irrigation requirements by directly
feeding local vegetation. This may facilitate the use of desert vegetation within composite ABS systems, containing a mixture of black panels, PV and vegetation.

## 3.2 Processes

To increase our process understanding of impacts we examined important convective processes - vertical motion and convergence, to assess the characteristics of these phenomena at different ABS scales. We present the analysis for the 27 July
which had a moderate-to-strong rainfall impact. As surface fluxes are dependent on land-atmosphere feedbacks, we also investigate how these feedbacks change using an extension of the Heated Condensation Framework index (Tawfik et al. 2015a; 2015b). A further goal is to assess whether impacts can be predicted, in terms of necessary surface heating and the hypothetical time it would take to produce this heating over ABS.

Over each ABS, the albedo was prescribed in the WRF model static data as 0.05. This contrasts with the albedo of surrounding
desert soils which is around 0.3. As a result, net-shortwave radiation absorbed at the surface is increased by 25% of the incident solar radiation, and this extra energy is partitioned into turbulent and ground heat fluxes. As there is virtually no available soil moisture or vegetation in the UAE lowland deserts, the latent heat fluxes are negligible, and the energy is partitioned almost entirely into sensible heating and ground heat flux. Figure 7 (left axis) shows the mean and standard deviation of sensible heating (areal mean) over a common 50 km footprint (left panel) as well as the individual ABS footprints (right). Firstly, the
50 km footprint average allows for a consistent comparison, but it means that some desert is included in averaging for the smaller ABS. It gives a sense of how the different scales impact on atmospheric heating over a large area, in comparison to control. Second, averaging over the individual footprints provides insight into how the different scales change the feedbacks determining the sensible heat flu, e.g., due to influence from the surroundings.





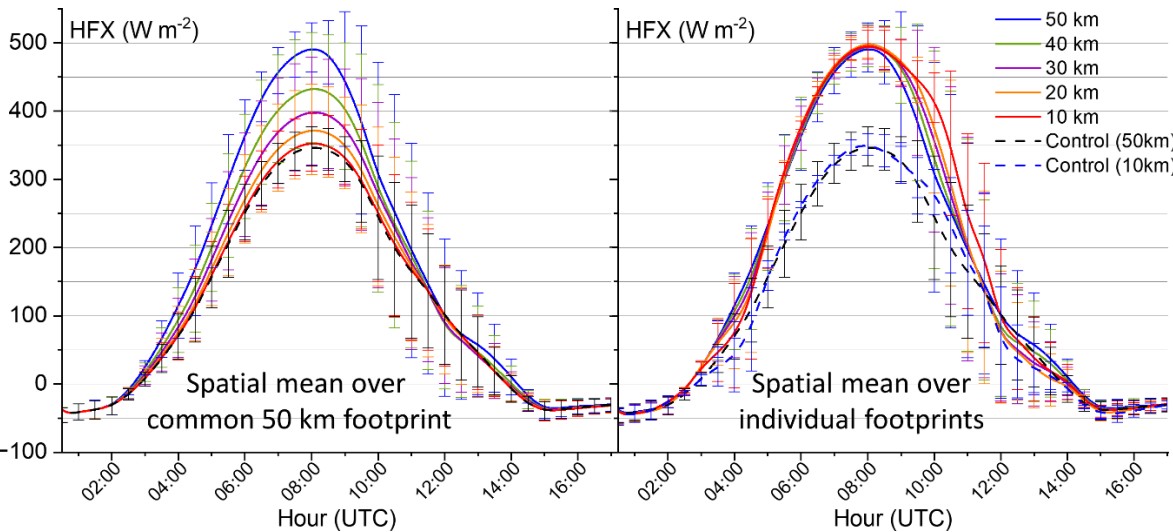

**Figure 7: Mean sensible heat flux and standard deviation on 27 July inside perimeter of both 50 km ABSs (left) and over each individual ABS footprint (right). In the right panel, the averaging for Control is shown for both the 10km and 50 km footprint.**

In both panels, there was a night-time downward flux of sensible heat, when nocturnal air temperatures were still high and the ground was cool. At around 07:00 local time, upward sensible heating flux increased rapidly until 12:00 local time. In the left panel, there is ~150 W m$^{-2}$ difference between the Control and 50 km ABS, with the difference between the respective peaks decreasing non-linearly with scale. The 10 km ABS curve differs little from Control, which is not surprising given that most land cover in the 50 km zone is desert. In the right panel, differences are subtle but perhaps informative. In the late morning and at midday, the average heat fluxes are similar when averaged over the ABS areas, but between 03:00 and 05:00 UTC (07:00 and 09:00 local time), the average heat fluxes diverge. This indicates either a) advection effects in respect to ABS scale and downwind fetch effects, b) scale-dependency of PBL development and free-troposphere entrainment, or simply c) spatially-varying atmospheric conditions. Regarding the influence from the surroundings, either at the surface or above, one may expect the 10 km ABS to be most affected, given its smaller size. The average heat flux is ~40 W lower than over the 50 km scale during this time (right panel) which indicates that this may be the case. However, mean diurnal values do not necessarily tell us why this is the case. The variability is also positively scale-dependent at this time, as evident from the error bars. However, it is not completely clear if this simply due to greater HFX values in the larger scales, or whether this is due to land coverage and the number of grid cells in each scenario.

Figure 8 shows us the spatial distribution of heat flux and CIN at the pre-convective time of 07:30 UTC. In the top row (panels a-f), we can see higher values of HFX on the southern windward side of the 50 km ABS (panel b) than for the lower scales (c-f). However, the patterns of the HFX gradients are quite similar across all scenarios, as compared with the Control (a), indicating that the 50 km ABS has higher HFX values simply because it extends further south into an area where atmospheric conditions produce higher HFX values. This idea is evidenced further by the similar spatial pattern of HFX in Control. The HFX magnitudes are different but the spatial distribution is similar. Differences in static land surface characteristics, such as





soil texture or moisture, were considered as possible reasons for these patterns but these were disregarded. The soil moisture
is virtually zero and the soil texture is very homogenous over the whole area. Hence, the reasons are likely due to atmospheric

conditions. However, as well as the surface flux, we investigated impacts higher in the atmosphere. The bottom row shows
convective inhibition (CIN, as also seen in Figure 3) at 07:30. CIN is the energy required for a hypothetical surface parcel to
reach the LCL and the LFC, and is computed as the area bounded by a parcel trajectory from the virtual temperature profile
(e.g., in a Skew-T plot). Thus, it characterizes differences in integrated temperature profiles. Panels (g-l) show that, although
the HFX spatial patterns look similar, there are clear differences downwind in the way that the HFX breaks down CIN, through

surface heating. This is likely due to the time needed for surface winds to advect across the ABS. There is simply more time
for heat to be transferred upward with the most heat accumulating downwind. In the 10 km ABS (panel l) the CIN only breaks
down to 80-100 W m$^{-2}$, 20 km reaches 50-60 W m$^{-2}$ (k), and at the extreme 50 km ABS (g) CIN is lowered to 0-10 W m$^{-2}$. Of
course, CIN represents the thermodynamic environment only, and dynamic effects, such as sea breeze fronts, can play a large
role in CI e.g., through lifting.

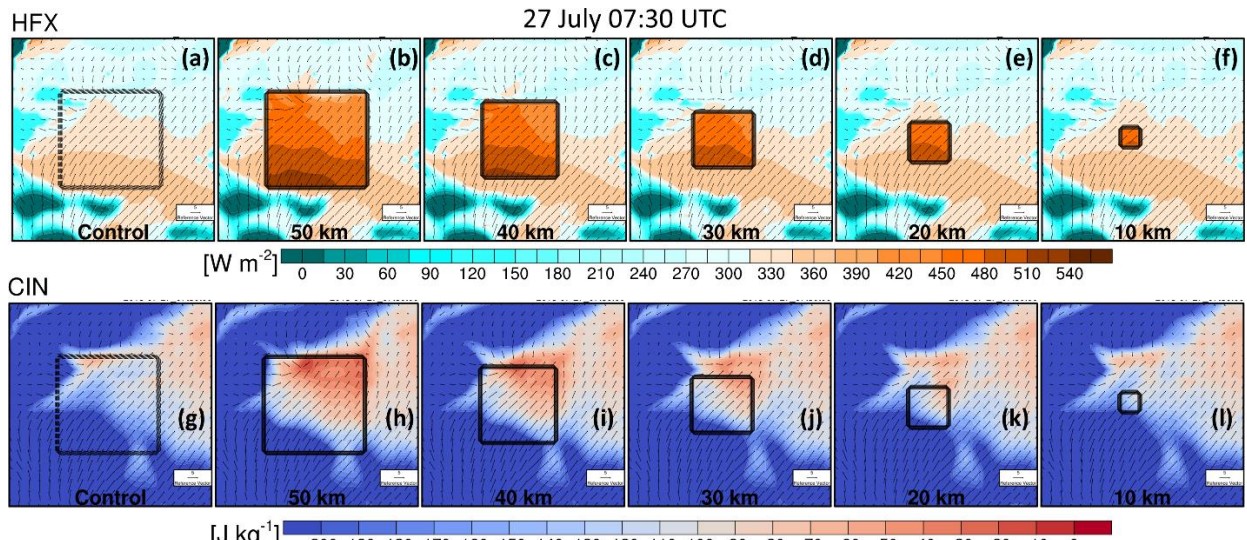


**Figure 8: Sensible heat flux (top row, W m$^{-2}$) and Convective inhibition [CIN] (bottom row, J kg$^{-1}$) at 07:30 UTC on 27 July.**

The increased heating and buoyancy seen in Figure 8 is likely to produce stronger vertical motions and convergence lines –
the latter especially when landscape discontinuities lead to differential heating. Figure 9 shows surface convergence (vertical
mean 0-500 m AGL), 10 m wind vectors, and 2 m temperature (on 27 July). The printed values are mean convergence values

within the 50 km zones. To the west in the Control (panel a), there are convergence lines below the (hypothetical) ABS
stretching diagonally from left to right (panel (a)), and the location of these corresponds with the light Control rainfall on this
day (Figure 4). The convergences align approximately with the 10 m wind direction, and the wind vectors either side are
deflected toward them. If we compare with the 50 km scenario (b), we can see that the temperatures are increased (wind vectors





are redder in color), and the convergence lines are intensified. As we go down in scale this intensification decreases, and this

is reflected in decreasing mean values.

Impacts on the eastern side are even more interesting (where the rainfall impact was larger), as there is a stronger difference in convergence impacts. Here, the Control has only weak convergence (caused by south-westerly winds meeting the light sea breeze), but these are strongly intensified with the ABS, which produce new convergence lines. The dependence on scale is clear with strong upward motion evident in the 50 and 40 km ABS (b and c), which gets progressively weaker at lower scales.

At 10 km (f) there is little difference when compared with the Control (a). Hence pre-existing convergence lines (Control) may be amplified by an added ABS.

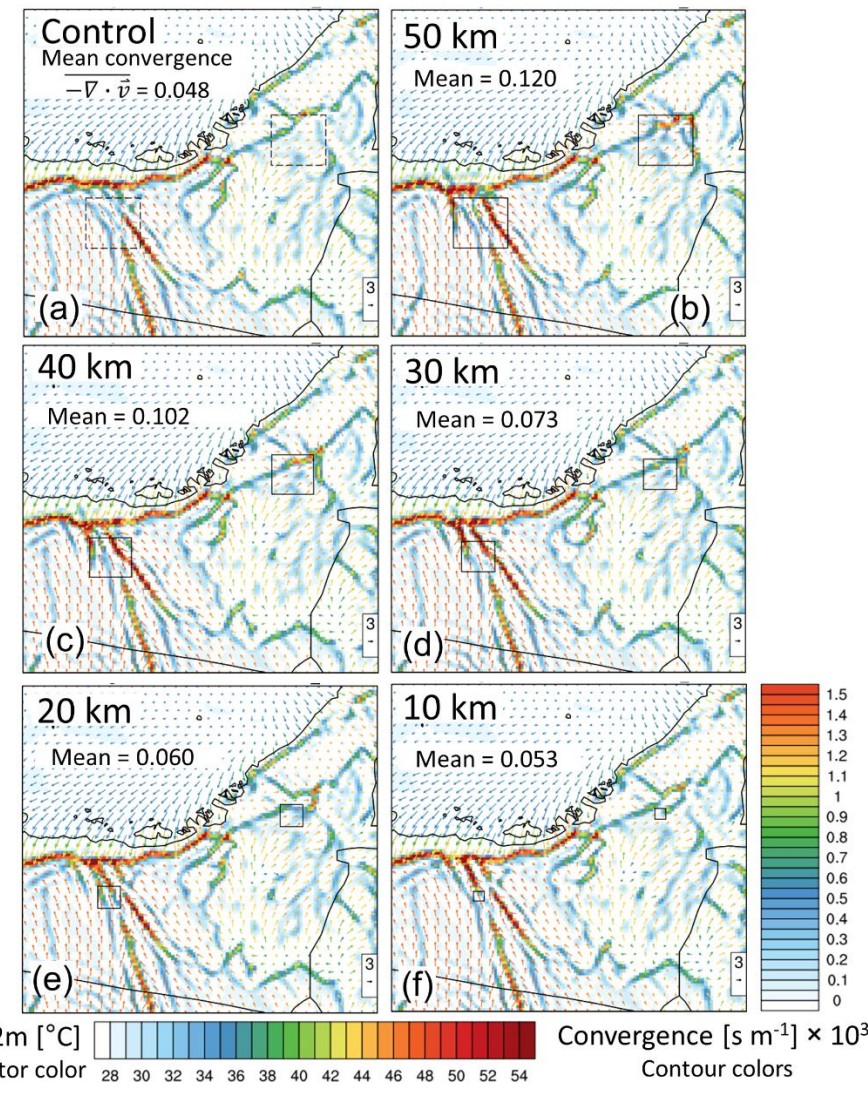

**Figure 9: 2D images of pre-convective convergence (mean over 0-500 m AGL), 10m wind vectors and 2m temperatures (vector colors) at 09:00 UTC on 27 July. The text values are mean convergence over the 50 km footprint.**





Figure 10 shows time-height diurnal plots, to see how the convergence field develops over the day up to 6 km in altitude, potential temperature (contour lines) and the average height of the PBL ($Z_i$, thick solid line) and the height of the level of free convection (LFC, dotted line). Three ABS sizes are shown (Control, and 50, 30, 10 km). In all scenarios (panels a-d), there is a low-level mean convergence and a corresponding divergence above ~2 km height (at approximately the time of convection, 09:00 to 09:30). However, differences are visible between the scales. In the 50 km ABS (c), both convergence and divergence

are strongly increased. Furthermore, the LFC is lowered and $Z_i$ is increased, almost meeting the LFC (in fact the LFC is breached in various grid cells). There is also greater de-stratification of potential temperature due to more vigorous mixing, as indicated by the isolines in the lower center of panel (c). Again, these effects tend to decrease with descending scale (b and d), and indeed the 10 km scenario appears indistinguishable from Control. Interestingly in the 50 km scenario, low-level convergence and PBL development are impacted earlier in the day, indicating that diurnal timing may also be important for

CI.

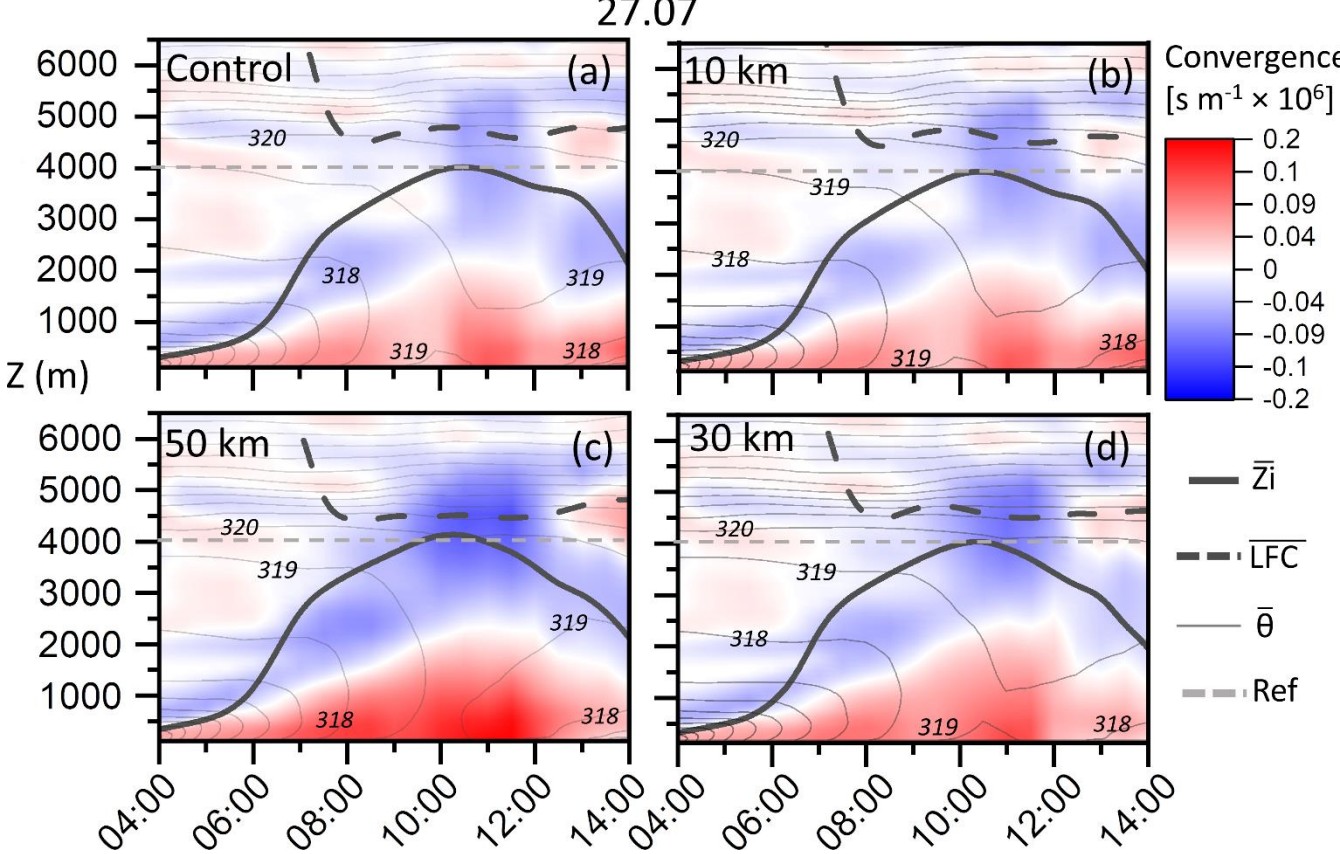

**Figure 10: Time-height vertical plots of the mean convergence over the 50km footprint (blue-red contours) on 27 July. For clarity, only four of the scenarios are shown. Also shown are mean boundary layer heights (Zi) and mean level of free convection (LFC). Zi is calculated using the gradient Richardson number with a critical value of 0.25 (Lee & De Wekker (2016); Seidel et al. (2012).**
**grey contour lines show the mean potential temperature (Θ, K), and the grey line is a 4000 m reference.**



Convergence goes hand-in-hand with mean vertical velocity (w). In Figure 11, updrafts (from inside a common 50 km perimeter) were plotted against height at 08:00 UTC (27 July), along with a line of best fit. From this figure, we can compare four important aspects: a) maximum velocities, b) the height at which w maxima occur (righthand peak), c) the maximum height at which any updrafts occur, and finally d) the total number of updrafts (bar chart).

For aspect (a), the peaks indicate a positive but non-linear relationship between scale and w maxima reached, with maximum values increasing in a somewhat exponential manner. For aspect (b), we see that with increasing scale the height of the peak where the strongest updrafts occur also increases (righthand peak of each set of dots). At 10 km, the peak does not rise above 1200 m. At 20 km it occurs at >1500 m and at 50 km >2000 m. From the best-fit lines, we can see that the slope angle changes from negative to positive as we step up from 10 to the 20 km scale, indicating a transition in the way vertical updrafts are

constrained by scale. Regarding aspect (c), the height at which updrafts occur (in this case up to 4500 m) is also influenced by scale. At 20 km, updrafts exist up to 4000 m high, and from the 30 km scale upward they reach >4250 m. This is important, because the mean LFC at this time (08:00) is ~4300 m height (Figure 10, dotted line). At the 10 km scale, there are no updrafts at all above 3500 m, which also provides a subtle transition as this is well below the average LFC height. Regarding aspect (d), we see a positive relationship between scale and the number of updrafts. This appears to be a fairly linear relationship

from 50 down to 20 km. Below that, the number of updrafts drops quickly, which also points to a transition between 10 and 20 km.

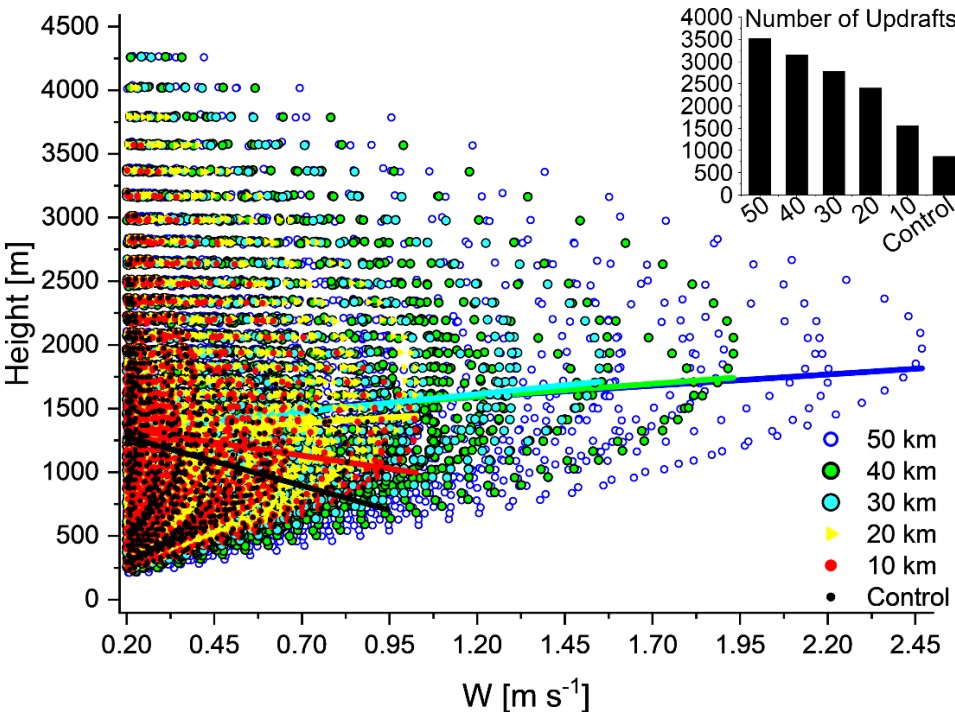

**Figure 11: All vertical velocity values (0-4.5 km) and their heights on 27 July at 08:00 UTC within the 50 km footprint. A line of best fit for each scenario is also plotted to show the relationship of velocities with height. The bar chart shows the total number of updrafts**
**for each scenario (within the 50 km zone) at 08:00 UTC.**



## 3.3 Required Heating Rates for Convection

Our goal is to gain insights into how scale influences convective impacts, and it is known that CI is shaped not only by large-scale conditions, but also by land-atmosphere (LA) interactions, or feedbacks (e.g. Jach et al., 2020, Gerken et al., 2019). The relative impacts of LA feedbacks increase in weakly-forced large-scale conditions. LA feedbacks can be defined as co-variabilities between land and atmospheric states, with the timescales spanning from minutes (surface heating - surface fluxes) to years (soil-moisture - precipitation). LA feedbacks are often quantified using process or statistically-based LA feedback indices (Santanello et al., 2019).

Surface heating is the dominant process behind artificial heat island impacts (Branch and Wulfmeyer, 2019), and our goal is to investigate what spatial scale and level of heating is likely to produce convective impacts in the UAE summertime. If the surface heating rate required for convection can be quantified, we can estimate how long it would take to reach CI for a given heating rate –a 'convective timescale'. To do this, we applied the Heated Condensation Framework (HCF) index (Tawfik et al., 2015a; 2015b), which takes an iterative surface-heating perturbation approach, following a four-step computation:

- Increase the near-surface potential temperature ($\theta 2m$) by a small increment, $\Delta\theta$.
- Find the height where the near-surface parcel ($\theta 2m + \Delta\theta$) is neutrally buoyant.
- Mix the specific humidity profile from the surface to the level of neutral buoyancy, therefore returning a mixed layer with constant humidity.
- Check if saturation occurs at the top of the potential mixed level by comparing humidity and
- saturation specific humidity.
- Repeat until saturation occurs (see Tawfik et al. 2015a for more details).

The index provides, amongst other quantities, a buoyant convective level height, or BCLH (that the PBL must reach to initiate convection), and a sensible heat deficit (SHDEF) required for CI to occur based on the atmospheric state. One advantage over indices like CAPE, according to Tawfik at al., is that it avoids pitfalls associated with parcel theory, because it is dependent not only on the properties of a hypothetical surface air parcel, but is a property of the air column as a whole. The time evolution of sensible heat flux deficit (SHDEF, GJ m$^{-2}$) for the 27 July is shown in Figure 12 – averaged over the eastern ABS. The largest differences in SHDEF occurs in the two hours between 05:30 and 07:30. As one might expect, the deficit is lowest in the 50 km ABS, and the deficit increases as the scale decreases. In the different scenarios, equal values of SHDEF are often reached, but at different times. For example, at 50 km the deficit falls to 6 GJ m$^{-2}$ before 06:00, whereas Control only reaches this deficit later, between 06:30 and 07:00. Large-scale conditions can change over the course of one day, e.g., the developing heat-low over the Arabian Peninsula, so the timing of surface heating may be just as important a factor for CI as heating strength.



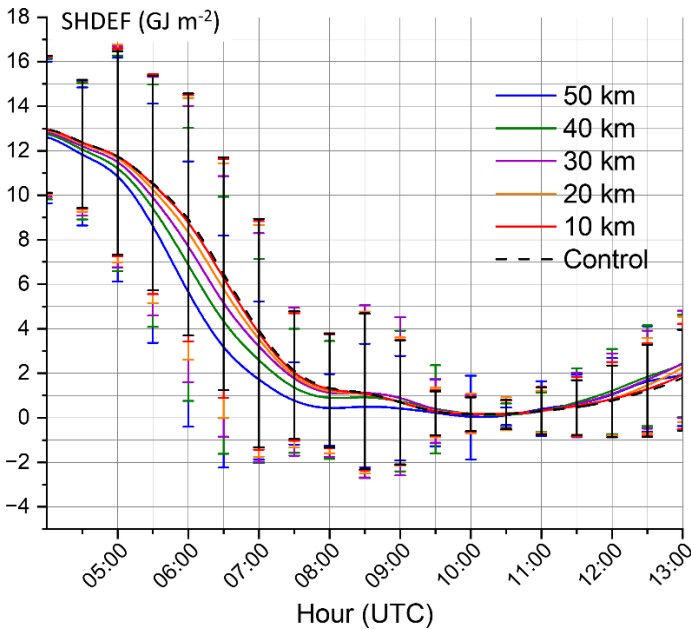

**Figure 12: Mean sensible heat deficit (SHDEF) on the 27 July over the eastern 50 km ABS. Also shown is the spatial standard deviation.**

We then applied the heating deficit (SHDEF) to calculate convective timescale ($T_{CI}$), or the time required to reach CI (minutes),

based on the actual sensible heating rate (HFX, W m$^{-2}$). This, was calculated in Eq. (2) simply as:

$$T_{CI} = \frac{SHDEF}{HFX \cdot 60s},$$  (2)

The convective timescales are plotted in Figure 13 and 14, this time for all cases. Figure 13 shows $T_{CI}$ at 06:30, roughly 2.5 to

3 hours before the onset of rainfall (09:00 - 09:30, Figure 5), and Figure 14 is the equivalent, half an hour later. From Figure

13 we can see that the 18 July has almost no $T_{CI}$ values below 420 mins at this time, and this may be the reason why the rainfall

impact is weaker, and starts later than the other cases. In the Skew-T there is a dry layer above the surface (Figure 2, panel

(a)), in which we would expect dry-air entrainment to reduce buoyancy substantially. This dry layer would be reflected in the

HCF computation which mixes column humidity before checking for saturation.  In all cases, the Control scenario drops no

lower than ~100 minutes, which may be too long for surface parcels in a dry environment to reach the LFC – which is ~4000-

4500 m at this time of day (Figure 10, panel (a)). At the 10 km scale, the mean timescale (values above panels) required for

CI is hardly reduced when compared to the Control (15-30 mins), apart from on the 18 July where there is a 100 min reduction.

At 20 km, where there is generally a more sudden decrease in the time required. There are grid cells in the 10 km scale where

short timescales are reached but the spatial extent of these areas are small and this may inhibit the possibility of CI.

If we compare the same index one hour later (Figure 14), we can see that all ABS scales are now at or below a 10-minute

convective time, indicating higher potential for impacts. Even at this later time the Control still does not reach short convective

400  times (over the 50 km ABS area), but there are some areas around this area with shorter times, indicating instability and this

is reflected by the rainfall patterns (Figure 4).  Impacts are very likely constrained by two factors - the horizontal extent of the





ABS and the later 'onset' time for shorter convective times, e.g., <10 min. As before, we may deduce that the time when $T_{CI}$ approaches zero is likely a big factor. The two days with the most rainfall, the 25 July and the 27 July, exhibit short timescales earlier and over a larger area of the ABS, than the other cases (Figure 14, bottom two rows).

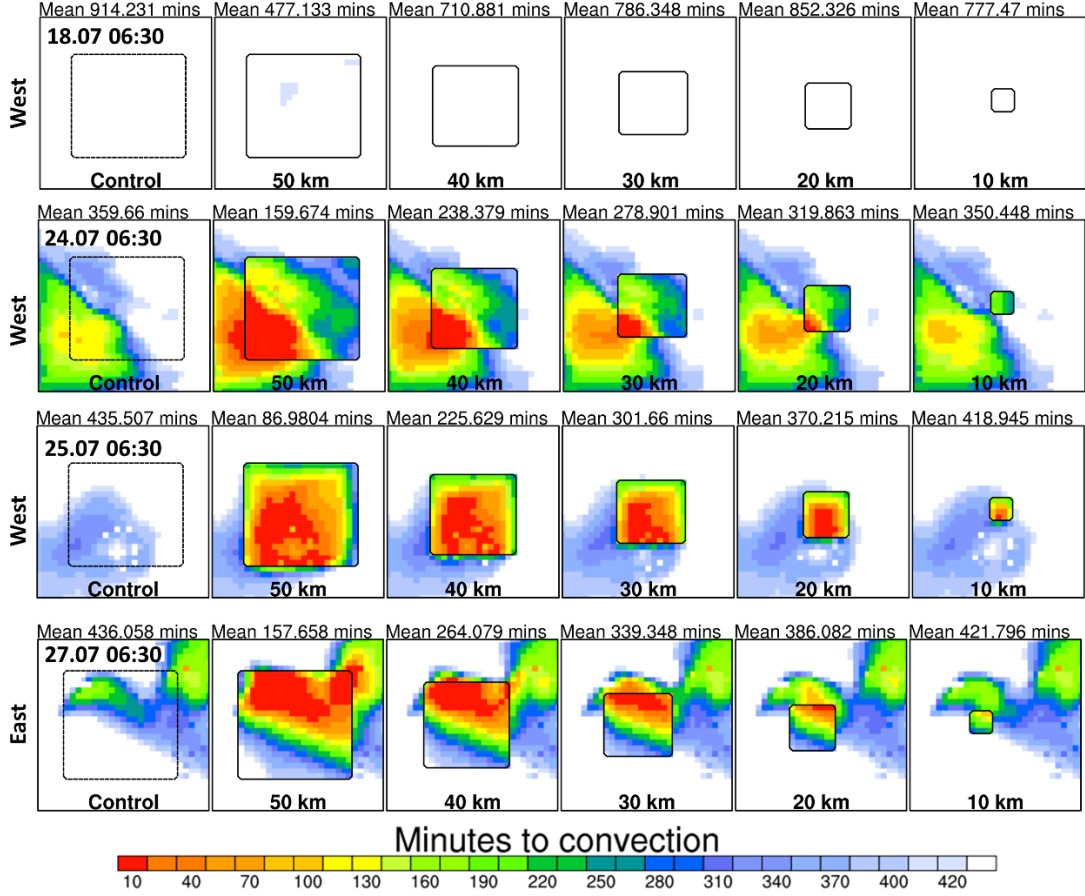

**Figure 13: "Time until convection initiation" for each grid cell based on the Heated Condensation Framework *(Tawfik et al., 2015a)* at 06:30 UTC**



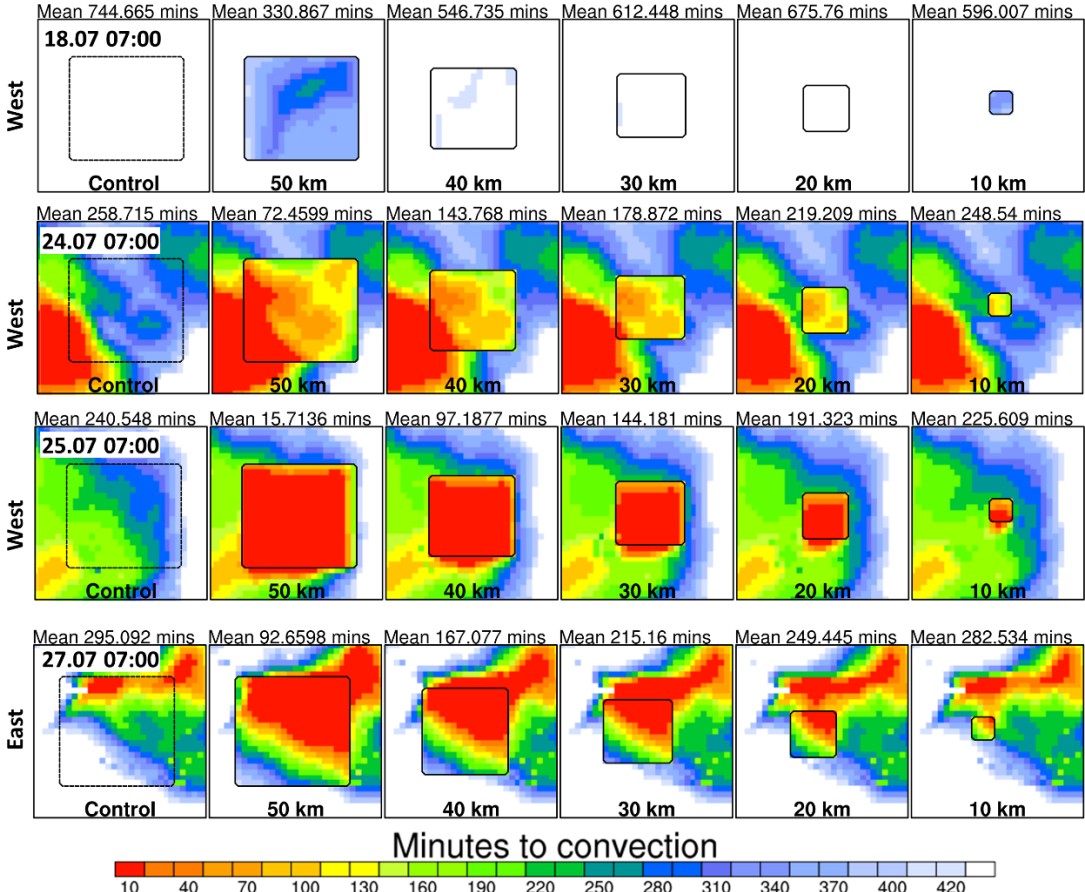

**Figure 14: As for Figure 13, but half an hour later at 07:00.**

Figure 15 shows areally-averaged $T_{CI}$ within the respective east or west 50 km perimeters. Between cases, there are differences in the appearance of the curves in the 40 and 50 km scenarios, both in slope, time, and magnitude. In general, onset times are earlier in the 24 July and 25 July cases (panel b and c) and the timescales both drop to zero minutes. On the 18 July (a), mean $T_{CI}$ does not fall much below 45 minutes (except for the 50 km scale) reflecting lower impacts in general, and the onset times (when $T_{CI}$ values drop rapidly) are later on than in the other cases. The 27 July (d) has lower $T_{CI}$ values, and earlier onset times than 18 July, but later than 24 and 25 July. In general, for all cases, the spacing between the lines increases with scale. What is also interesting is that there are differences in the slope from high-to-low $T_{CI}$. The strongest cases (24 and 25 July) tend have steeper slopes which occur early. The 25 July where the strongest impacts occurred has the steepest slopes from all of the cases, which hints that the rate of change of HFX is also an important factor.



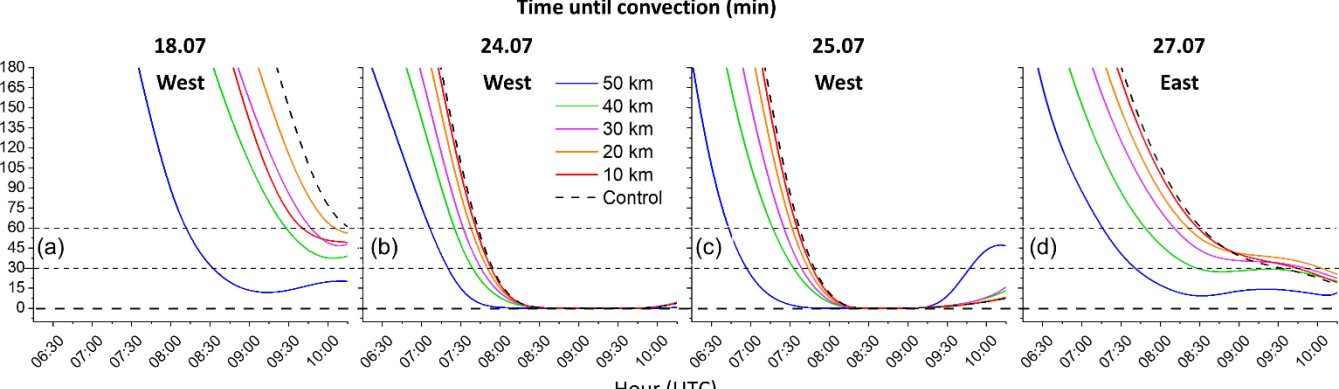

**Figure 15:** "Time until convection" based on the HCF index ($T_{conv}$, mins) - areal-mean over a 50 km footprint - cases are split into east or west areas, based on strongest rainfall impacts.

# 4    Summary and Outlook

In this study, we investigated whether artificial black surfaces can enhance rainfall in the UAE, taking a validated model approach. Our objectives are to examine the scale-dependency of ABS impacts in respect to rainfall enhancement and convective processes, and to explore the predictability of impacts by applying the LA feedback metric, the HCF index.

Simulations of four moderately-unstable cases demonstrate that rainfall is enhanced in the UAE summertime by low-albedo surfaces, primarily through intense surface heating. At this time of the year, there is a daily sea breeze which brings humidity, convergence lines and reduced CIN along the front, but this is not always sufficient to trigger CI. The addition of large ABS acts to amplify these factors by pre-conditioning the atmosphere, further reducing CIN, and intensification or creation of convergence lines.

We have demonstrated that ABS scale is important, and that impacts appear to increase proportionally with area in respect to rainfall and convective processes like convergence and vertical accelerations. Specifically, there are scale-dependencies on production of surface convergence lines (0-500 m), mean convergence (0 - 2 km); maximum vertical velocities, the frequency and height of updrafts, and modification of PBL and LFC height. Additionally, aspects of the convective-timescale curve (Figure 15) appear to be related to impact strength - specifically the daily timing (onset) and the downward rate of change. If the convective time dip starts earlier in the day for instance, the impacts are likely to be stronger, because more pre-conditioning occurs before the sea breeze front arrives – which usually arrives at a similar time of day. Additionally, a steeper slope seems to be associated with stronger impacts. The steepness could be increased by influxes of humidity, a more unstable atmospheric profile, greater solar radiation intensity, or other factors, all of which would reduce the sensible heat deficit, as derived from the HCF. Both the convective time onset and slope is likely to vary from day-to-day with weather conditions, as well as with ABS scale.

As well as identifying relationships between impact and scale, we have also gained insights into lower scale-boundaries in the 2015 summer. At ABS scales of 10 km, sensible heating rates (per unit area) are relatively similar to 20 and 30 km scales



(Figure 7 and 8), but there is little effect on convective processes and rainfall, whereas from 20 km upwards, impacts occurred in all cases. The lack of impact at 10 km is due primarily to insufficient heating over a wide-enough area. The smaller footprint and mean that an advected air mass does not accumulate enough heat to reduce leeside CIN (Figure 8), and thus the near-surface remains stable. Furthermore, compared to larger scales, the 10 km ABS does not produce strong updrafts, and especially not at heights which approach the LFC (Figure 11). In fact, the velocity-height trend line demarcates the 10-20 km

transition quite well and may provide a useful metric for further prediction analyses. Furthermore, the convective timescale, has provided useful insights into impacts of scale and onset timing, and could be used with regional climate simulations to predict ABS impacts within a given climate envelope.

In terms of rainfall enhancement, if we assume that the water quantities produced by the model microphysics are plausible, then the implications are considerable. Extrapolating from the amounts, if the UAE implemented just one pair of ABS surfaces,

of 20 or 50 km scales, then ten rainfall enhancement 'events' per year would supply enough water for 3000 or 15,000 people. These amounts would increase if more events occur, or if more than two such ABS were implemented.

Confidence in rainfall enhancement should be further tested in further studies which assess simulation sensitivity, regional climate variability, and statistical analyses. Ensemble UAE simulations with varying model physics including microphysics (e.g., Schwitalla et al., 2020; Fonseca et al., 2020, Taraphdar et al., 2021) would be particularly useful. Here, data assimilation,

quantitative precipitation estimation, and rain radar analyses may also be employed (Bauer et al., 2015; Kawabata et al., 2018; Branch et al., 2020; Schwitalla & Wulfmeyer, 2014). Sensitivity to model resolution could be investigated by conducting large eddy (100-300 m) or turbulence-permitting (< 100 m) simulations (e.g., Bauer et al., 2020; 2023). A caveat here is that evaluation of model microphysics at finer scales remains scarce, unlike convection-permitting (CP) simulations (2-4 km grid-scale). Fine-scale simulations would certainly provide further insights into physical processes. Finally, long-term temporal

variability of impacts may be tested further by application of climate or multi-seasonal simulations, together with statistical analyses.

In summary, enhancing freshwater resources is essential for countries like the UAE, where water scarcity is becoming more severe, especially under continuing climate change. ABS systems offer a means of combatting this scarcity, augmenting the UAE's strategy of cloud seeding, desalination, and water importation. ABS could be composed of a combination of mesh, PV

panels and plant canopies, as long as the net effect of modified albedo, roughness and energy balances are conducive to enhancement. Large-scale solar farms are being realized in the region already, and through clever design, there may be an opportunity to take advantage of combined rainfall enhancement and power generation. For instance, one may switch off solar electricity generation for a short time, when conducive atmospheric conditions are present. This would increase the heating of the atmosphere during the best conditions for rainfall enhancement. If a balance between baseload requirements and rainfall

enhancement can be struck, there could be great potential for this synergy. There are other synergies too, such as the possibility to use solar-powered irrigation systems or desalination. Thus, ABS systems offer a flexible means of covering large areas to enhance rainfall and should therefore be made a high priority for further research.



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

**Code availability**

The WRF V4.2.1 model code and all relevant information is available at https://www2.mmm.ucar.edu/wrf/users/

Data analyses was done with NCAR Command Language (NCL) V 6.6.2 software (2019)

Bounder, Colorado: UCAR/NCAR/CISL/TDD. Available at http://dx.doi.org/10.5065/D6WD3XH5

Other graphics were produced with OriginLab V10.0.0.154 (Academic) Software (OriginLab, Northampton, MA). Available at https://www.originlab.com/

**Data availability**

The WRF model output data created for this study is too large to store in standard repositories, but is available in netcdf format on request.

Access to the GRIB ECMWF Operational Analysis model level data used for forcing WRF is restricted and not freely available. Further information may be obtained at https://www.ecmwf.int/en/forecasts/dataset/operational-archivev

**Author contribution**

Oliver Branch conducted all simulations and the majority of data analysis and preparation of the manuscript

Lisa Jach provided scientific support and assisted with analysis and manuscript preparation.

Thomas Schwitalla provided scientific support and assisted with analysis and manuscript preparation.

Kirsten Warrach-Sagi provided scientific support and helped to prepare the manuscript

Volker Wulfmeyer contributed to experimental design, scientific support and helped to prepare the manuscript





**Competing interests**

The authors declare that they have no conflict of interest.

**Acknowledgements**

This material is based on work supported by the UAE Research Program for Rain Enhancement Science, under the National Center of Meteorology, Abu Dhabi, UAE.

Furthermore, we are grateful to the High Performance Computing Center, Stuttgart (HLRS) for providing support and

computing time on the Hawk Apollo system.

We are also grateful to ECMWF for providing operational analysis data.