# Peer review of "Scaling Artificial Heat Islands to Enhance Precipitation in the United Arab Emirates"

_EGUsphere, 2023_

## Referee Comment (RC2)

Review of Branch et al (2023)
*Scaling Artificial Heat Islands to Enhance Precipitation in Arid Regions*

The authors present a study of the impact of artificial reductions in surface albedo as a means to enhance convective precipitation over the hyper-arid UAE in a convection-permitting modeling framework. These reductions in surface albedo are imposed as uniform changes to the land surface albedo over a prescribed area in the model, but in reality would be achieved through some combination of solar panels, vegetation plantations, and other artificial surfaces that are much darker than the surrounding desert. Fundamentally, these changes in albedo impact precipitation in the model through their enhancements of surface sensible heat flux, which in turn has a variety of meteorological consequences. The sensitivity of precipitation impacts to an increasing area of albedo modification is also a key consideration.

I found the study to be reasonably well constructed and executed. The modeling tools and analysis seem appropriate for the key questions of the work. It's an interesting and relevant topic of study, and I am curious how this work will continue or even be applied in the future.

**Comments**

I believe the paper could be improved through an expansion of the context provided first in the Background section, which the authors then return to in the Summary and Outlook. For example,

- Line 33: The authors reference a few papers presumably showing that albedo "can trigger regional scale impacts," but they don't provide any detail on what those previous papers were studying and whether they are relevant to this paper in particular. If you are just trying to generally say that albedo can alter regional climate and weather, I'd also look for a bit more detail here highlighting some of the ways that people have shown albedo changes altering regional climate and weather. But it would be especially good if they were directly relevant to this work and connected into the introduction more smoothly.
- Lines 40 - 45: Branch and Wulfmeyer (2019) seems very relevant to your work here, and I would like to see a bit more drawn from that paper into your introduction in terms of what they found, how your approaches are similar/different, how it might have motivated this study etc…

I also believe the paper could benefit from more detail on prior work looking at convection & precipitation and its diurnal cycle in the region more broadly, at least during the summertime months considered. How often does it occur? Does it go right to deep precipitating convection or is there shallow convection first? What's the variability? Are there any consistent patterns, and when might those patterns break down? I think this would feed well into your proposed future work directions at the end of the paper, expanding into other times of year or if there's climate variability. It would just be helpful to know more about what the background setting of convective precip is in the region you're looking at, in case the reader is less familiar. In other words, what exactly are we modifying through the albedo perturbation in the first place!

Similarly, I think the authors need to provide a bit more detail on how the model they are using does with convection and precipitation in this region. They note a validation study in Line 90 from a few years ago, but they don't indicate whether the results of that evaluation were favorable, especially for the variables they're pulling out of WRF in this analysis. Does the model reproduce CAPE over the UAE well compared to soundings, for example? Could you show satellite imagery, if radar isn't available, showing that your control simulation produced reasonable patterns convection on your four case days?

This is also important to discuss in your Model Configuration section if the only prior validation was done against surface observations. And along those lines – I would be careful in saying that this version of WRF is the same as the "validated" version when the "only change" was to use an updated version of the PBL, surface layer, and land schemes (Line 92) – the components of the model that are among the most important for this study. Just because a model component has been updated doesn't guarantee that it will improve the skill, especially over a particular region and when looking at something as sensitive as convective precip. Overall, I would just be more clear about what has been and what hasn't been validated for this particular region using this configuration of WRF. And it may be the case that some aspects (like my CAPE example above) haven't been tested exactly, but I would just mention that as a caveat and/or an area for future work.

Additionally, I think you have an opportunity to better connect your work to other areas of surface-atmosphere interactions in the Background (Lines 60 - 72) and the Outlook to round out the manuscript. Enhancements in sensible heating, usually associated with changes in vegetation cover or properties, have been shown to alter cloudiness/convection/precip in different ways depending on where you look. Your work highlights the importance of background humidity (here brought by the daily sea breeze) that can then be lofted to saturation by deeper, more vigorous boundary layers, which is something that also comes up over vegetated surfaces. I think highlighting that similarity and connection, especially in an environment with little-to-no latent heat flux would ground the study in prior/ongoing work more completely.

My final "general comment" deals with the temperature effect of the albedo perturbation introduced by the ABS. This study is mainly looking at precipitation effects, which makes sense given the region, but I would also look for some discussion of other implications of this strategy. If we darken the surface, we will also increase near-surface air temperatures. Will this be a problem locally for people, even if it is helping with some of their water scarcity issues? Maybe the temperature change isn't impactful relative to the background hot climate. If the area of the albedo perturbation gets large enough or there are too many of them, could it alter sea breeze or other regional dynamics in unhelpful (or maybe helpful) ways? If this is being considered in the context of a regional climate strategy those other effects need to be mentioned, at least. And again, that might be an area of future work for this manuscript, but I do think it needs to be mentioned.

Minor/Detailed Comments:

- Line 47: what kinds of weather modification have the previous studies shown for the PV panels? This ties back into my general comment above about more detail in the background, but just wanted to make a specific note here. Also, I don't believe you ever spelled out PV as photovoltaic, which would be good for the first use.
- Line 57: somewhat tangential, but are those PV panel efficiencies valid for an environment like the UAE with such high temperatures?
- Line 70: found "interactions" between urban heat islands and sea breezes, particularly for convection  – what are those interactions? Are they relevant for your results here?
- Line 83: The term Artificial Heat Island is not used in the introduction. Given how it's used to frame the study in the paper title, I would look for it to show up here somewhere. Perhaps in the connection from Urban Heat Island.
- Line 105: Can you clarify that this data assimilation of soil moisture and temperature is happening in the ECMWF system you use for your boundary conditions? Or is this going into WRF directly?
- Line 136: Could you provide more detail on how the case days were selected within the year and if the days themselves are representative of diurnal patterns in the study area (I see that you noted a validation that summer 2015 was climatologically representative but not the days themselves)?
- Line 154: When you say conditions in the east vary more with the Gulf of Oman, can you be more specific?
- Line 177: Is the model overly drizzly?
- Caption for Figure 4 references a 150 km diameter, but in the text it is 90 km.
- Lines 230 - 240: How confident are you in the ability to operationally predict where precip enhancements needed to be captured for human use? Is the enhancement based on the ABS here falling in the right spot that it *could* be collected or directed to groundwater recharge? Would be important to note if this is being used to justify any sort of deployment/construction.
- Line 245: I'm curious why you didn't pick the case where the rainfall impact was strongest?
- Line 258: "heat flu" is missing the x for "flux"
- The caption on Figure 7 was a bit confusing in terms of the left panel having "both 50 km ABSs", when you're just noting that they have a common footprint
- Line 283: Why were these other factors disregarded?
- Line 324: "diurnal timing" of what?
- Line 352: CI referring to convective initiation or impacts?
- Line 426: Just to clarify, is the HCF index what you are using as your LA feedback metric?
- Line 453: *Are* the water quantities produced by the model microphysics plausible? This could tie back into my question about more detail about model validation/future work and the context of convective precip over the region.

---

## Author Comment (AC1)

**Point by point reply to Reviewer 1**

Review for "Scaling Artificial Heat Islands to Enhance Precipitation in Arid Regions"

**Opening Reviewer comment.**

*In their manuscript "Scaling Artificial Heat Islands to Enhance Precipitation in Arid Regions", the authors present analysis of several numerical simulations exploring the impact of decreased land surface albedo in squares of various sizes over two desert regions of the United Arab Emirates. They find that, when the imposed squares of low albedo land are larger than 20 km per side, there is a statistically significant increase in precipitation in the surrounding region. The simulations are conducted using the Weather Research and Forecasting model, with boundary conditions forced for four distinct weather events in the region. The methodology is appropriate for the aims of the study, and the topic is appropriate for GMD. I have a few addressable (but still important) concerns, outlined below, as well as several minor comments. Following revision, this manuscript would be a valuable addition to the literature.*

We would like to thank the reviewer for taking the time and effort to review this submission. Your constructive comments and positive appraisal are much appreciated. We would like to address your comments below, point by point:

**Major comments:**

**Line 225:** "*It is not linear, but more exponential in appearance – which is reasonable, given the areal size increments.*" **--- I wouldn't make this claim unless you back it up with analysis, and you'd want to do the analysis with area of imposed ABS, not width of imposed ABS square (e.g. the 20km box is 4x larger than the 10km box); IS the relationship non-linear with area of forcing? That isn't at all obvious to me.**

This is a good point and the authors agree that one has to be careful. The intention here was to make a qualitative observation to describe the non-linear bar spacing only (Fig 6c), not necessarily to quantify a definitive area/precipitation relationship in terms of processes. We understand that and it is difficult to assess whether the non-linear rainfall amounts relates directly with the areal increase, or could be due to other factors too e.g. a non-linear response to the downwind path length (fetch) of the ABS. To clarify our intent, add some new ideas, and to avoid making unwarranted claims we have added a caveat, modifying the text (L226-227) from:

*"The mean values indicate a clearer relationship between scale and surpluses. It is not linear, but more exponential in appearance – which is reasonable, given the areal size increments."*

to:

*"After averaging the amounts over the four cases, the precipitation amounts (and increases over Control) suggest a non-linear relationship between precipitation and ABS side length, i.e., differences between bars increase with successive increase in scale. Intuitively, this might be expected, given the exponential areal increase in our ABS scenarios, but other factors could also be influential in addition to the surface area e.g. the ABS path length in respect to prevailing wind direction."*

**Line 240: It seems necessary at this point in the analysis to discuss the \*temperature\* and \*heat stress\* impacts of the ABS - are these locally confined to the region of the ABS or do they extend regionally? (I wondered if perhaps this would come up later in the paper, but (a) it didn't and (b) this is where I felt like I wanted to see it addressed.)**

This is an excellent suggestion, and we have considered this very important regional impact. We had elected to avoid making the manuscript unnecessarily long. However, in view of your comment we suggest to add a few lines of text to clarify how the simulated surface temperatures may change outside of the ABS zones – based on our four cases. We suggest to supplement this with a figure below if the editor agrees to include it. We suggest to add the following text (at L241):

*"It is also important to consider the effect of surface heating on temperatures outside the ABS zones, as large increases in temperature could affect local citizens and vegetation. Figure X shows the difference (mean of the*

*four cases) in daily maximum and mean temperatures for the 50 km ABS scenario. Panel Xa indicates the mean maximum temperature difference during the daytime. Within the ABS zones the largest values are around one-degree Kelvin higher than in the Control. There is a temperature increase in the surrounding areas particularly around the eastern zone, but the differences are relatively limited, both in spatial extent and the increase (~0.2-0.3 K). Curiously, there are also some minor cooling effects to the south of the ABS. There is also a small mean daily temperature increase outside the ABS (panel b, ~0.2-0.3 K), but these areas are quite close to the ABS zones. These simulated values indicate that there is a slight temperature impact on the near-surroundings, but even at the largest ABS 50 km scale, this is simulated as low to moderate."*

[Figure]

*Figure X: The case-average impact on daily mean and daily maximum 2-m air temperatures from the 50km ABS. Computed respectively, as $\frac{1}{n}\sum_{1}^{n}\left(\overline{T2m}\ ABS_{50km} - \overline{T2m}\ Control\right)$ and $\frac{1}{n}\sum_{1}^{n}\left(T2m_{max}\ ABS_{50km} - T2m_{max}\ Control\right)$, where n is the number of cases. Panel (a) is the daily maximum 2-m temperature impact, and Panel (b) 24-hour daily mean impact.*

The authors hope that this figure and discussion of the temperature impacts will provide a more comprehensive picture for the reader, and perhaps provide some measure of reassurance about regional temperature impacts.

**Line 250: "*As a result, net-shortwave radiation absorbed at the surface is increased by 25% of the incident solar radiation, and this extra energy is partitioned into turbulent and ground heat fluxes.*" ... presumably also into heating the surface which leads to increased LW up. Maybe this is what you mean by "ground heat flux", but typically we think of "ground heat flux" as energy stored by the ground at each timestep, ie that doesn't have to be removed as LW, SH, or LH. But heating up the surface leads to higher LW out , which isn't a turbulent flux.**

The authors agree that this could be expressed more clearly and so we have modified the text (L250) from:

*"As a result, net-shortwave radiation absorbed at the surface is increased by 25% of the incident solar radiation, and this extra energy is partitioned into turbulent and ground heat fluxes."*

to:

*"As a result, an increase in net-shortwave radiation occurs (~25%), and an increase in the net-radiation which is partitioned into sensible and ground heat and latent (negligible) fluxes."*

The authors assume that by introducing the net-radiation here, that changes in all longwave radiation fluxes are inherently accounted for when discussing the total energy balance.

**Minor comments:**

**Line 16: what do you mean by "one-day cases over a 24 hour period"?**

The authors agree this could be expressed more clearly, so we have modified this line from:

*"Simulations of five square ABS of 10, 20, 30, 40, and 50 km sizes were made on four one-day cases over a 24-hour period."*

*To:*

*"Simulations of five square ABS of 10, 20, 30, 40, and 50 km sizes were made on four one-day cases, each for a period of 24-hours."*

The authors trust that this should be clearer to the reader now.

**Line 26: how much direct warming comes from the albedo change?**

This may be answered only partially within the results of this study, because we have modified not only the albedo, but also the surface height/roughness properties (shown in Table 2). Even if we had not modified the albedo, the latter change would also influence the feedbacks which shape sensible heating rates, 2-m and skin temperatures, and so on. As the prescribed ABS height (50cm) and roughness length parameter (5cm), do not deviate drastically from the bare desert soil parameters though, the effect of roughness is likely to be relatively small compared to that arising from albedo change. In that case, then new figure proposed above (on 2m temperature impacts) provides a 'reasonable' estimate of the albedo impact on 2m temperature over the ABS zones. Around ~1 °K mean increase in maximum daily temperature, and up to ~0.8 °K mean daily temperature (averaged over the four cases).

These temperature impacts are now described in L241-245 thus:

*"It is also important to consider the effect of surface heating on temperatures outside as well as inside the ABS zones, because large increases in temperature could affect local citizens and vegetation. Figure X shows the difference (case-average) in daily maximum and mean temperatures between the 50 km ABS scenario and the Control. For maximum daytime temperatures (panel a), there is a maximum temperature increase of up to ~1 °K, inside the 50km ABS zones when compared to Control. In the surrounding areas, there is a temperature increase particularly around the eastern zone, but the differences are relatively limited, both in spatial extent and the temperature increase (~0.2-0.3 °K). Curiously, there are also some minor cooling effects to the south of the ABS. For daily mean temperatures (panel b), there is an increase of up to ~0.8 °K inside the ABS zones. Outside the ABS, the largest increases are ~0.2-0.3 °K, but these areas are quite close to the ABS zones. These simulated values indicate that there is a slight temperature impact on the near-surroundings, but even at the largest ABS 50 km scale, this is simulated as low to moderate."*

**Line 34: specify with albedo \*brightening\* - will be clear to some but not all, especially since the bulk of the paper is about albedo \*darkening\***

As part of this review we have already changed the text (L31-35) to:

*"Examples are marine cloud seeding to reduce coral bleaching (Latham et al., 2014; Latham et al., 2013; Tollefson, 2021) and deliberate albedo management through agricultural landscape planning, and breeding of higher-albedo crops (Doughty et al., 2011; Kala et al., 2022; Ridgwell et al., 2009). Their general aim is to investigate the potential for regional cooling of temperatures. The deliberate **increase** of albedo falls under the geoengineering category of terrestrial solar radiation management (SRM). And although geoengineering is considered to be global in scale, regional actions may trigger regional impacts (Quaas et al., 2016; Seneviratne et al., 2018) such as reduction of temperatures (Kala & Hirsch, 2020), whilst at the same time contributing toward reduced global forcing (Carrer et al., 2018; Sieber et al., 2022).*

We hope that this modification has also addressed this comment by specifying a deliberate albedo increase.

**Line 40: ". Branch et al., 2014, measured albedos of 0.17 and 0.12 for jatropha and jojoba plants, and the surrounding desert ~0.3, leading to temperatures up to 4°C higher than the surrounding desert (see also Saaroni et al., 2004). This heating led to greater simulated cloud development and convection initiation (CI) (Branch & Wulfmeyer, 2019)." --- make clear in 2nd sentence you're no longer talking about measured or observed things, but rather a model simulation**

Thank you for pointing this out. We have changed this text (L40) accordingly from:

*"Branch et al., 2014, measured albedos of 0.17 and 0.12 for jatropha and jojoba plants, and the surrounding desert ~0.3, leading to temperatures up to 4°C higher than the surrounding desert (see also Saaroni et al., 2004). This heating led to greater simulated cloud development and convection initiation (CI) (Branch & Wulfmeyer, 2019)."*

*to:*

*"Branch et al., 2014, measured albedos of 0.17 and 0.12 for jatropha and jojoba plants, and the surrounding desert ~0.3, leading to temperatures up to 4°C higher than the surrounding desert (see also Saaroni et al., 2004). A subsequent model simulation of jojoba plantations reproduced similar differential heating, and an associated increase in cloud development and convection initiation (CI) (Branch & Wulfmeyer, 2019)."*

The authors hope this clarifies that these results come from two related, but separate publications.

**Lines 46/47: be consistent - earlier references to albedo would make this "0.05"**

The authors agree. To make it consistent, we have now modified the specification of albedo from % to a 0-1 parameter (L46-47)*:*

*"Panels could be coated with black paint with an albedo ~0.05 or even with specialist coatings with < 0.01 albedo (Theocharous et al., 2014)."*

**Line 47: PV - define (photovoltaic I assume)**

Agreed. This line (L47) is now changed to:

*"Other surfaces which may modify weather are **solar photovoltaic (PV) panels** (Li et al., 2018; Lu et al., 2021; Mostamandi et al., 2022)."*

**Line 51: not just sensible heating, but also high surface temperatures (increasing LW out of the surface through increased surface T)**

The authors agree this could be clearer: In a similar manner to the changes at L250 we have modified the lines

*"With PV, one must also account for radiation converted to electrical power, which for a given radiation flux would leave less energy for sensible heating."*

to:

*"With PV, one must also account for the amount of shortwave radiation converted to electrical power, which may lead to modified skin temperatures, longwave fluxes, and net radiation."*

**Line 51: "net total" - net total what? (I assume "net total SW absorbed that goes directly into the local surface energy budget" or something along those lines)**

Agreed. The authors have modified this line (L51) from:

*"The net total can be expressed as an effective albedo, here in Eq. (1):"*

*To:*

*"The net total 'loss' of shortwave radiation can be expressed as an effective albedo, here in Eq. (1):"*

**Line 52: Is this equation necessary / especially as an equation on its own line? It doesn't get used in the rest of the paper, and you don't actually simulate these combined albedo/energy uptake surfaces, so it is kind of distracting… You could put it in-line and emphasize that it is for background info only, and NOT what you're going to try to model here.**

The authors are happy to change this to an inline equation, if acceptable to the editor. We suggest to remove the equation number and change the line (L52) to:

*"The net total 'loss' of shortwave radiation can be expressed as an effective albedo, as $A_{eff} = \delta + \varepsilon$.*

**Line 54: "delta" - This is the "regular" surface albedo, correct?**

This is correct. The authors decided to use the definitions provided by the referenced author (Taha, 2013). However, to clarify that reflectivity is the surface albedo, we have changed the line (L54) to:

*"where $\delta$ and $\varepsilon$ are reflectivity (albedo) and conversion efficiency, respectively (Taha, 2013)."*

**Line 55: This comment is kind of long and the actual paper doesn't simulate solar panels taking energy "out" of the surface energy budget, so perhaps consider altering this text to emphasize that it is background information for the reader, and that you \*aren't\* going to be trying to capture this in this study! This discussion definitely sent me down a rabbit hole when you were describing your methods, until I realized in the results that this discussion isn't actually applicable to the actual simulations you ran.**

The authors agree that this section could be shortened a little since detailed simulations of PV panels are not the focus of the paper. Therefore, we have reduced the following lines (L54-56) from:

*"PV efficiency can theoretically reach ~46% in laboratory conditions (Allouhi et al., 2022), but in reality, is usually closer to 10-20%, with most radiation transformed into heating (Taha, 2013). This may be useful for rainfall enhancement, but can reduce cell efficiency and longevity (Dwivedi et al., 2020)."*

to:

*"PV efficiency is typically only ~10-20%, with much of the radiation transformed into heat (Taha, 2013), thereby offering potential for rainfall enhancement if implemented on a large scale."*

Secondly, based on your comment, we clarify that we are not explicitly aiming to investigate PV panels in detail, but introduce them only as a possibility for use as an ABS surface. We propose to clarify that we are not focussing on PV or vegetation by modifying the following lines (L57-60) from:

*"Assuming the lowest albedo of 0.04, and panel efficiency between 0.1 and 0.15, this would yield effective albedos of 0.14 and 0.19, i.e., similar to jojoba and jatropha. For simplicity in this study we will use an umbrella term 'Artificial Black Surfaces (ABS)' for these systems, whether they are made of black panels, PV or any composite of such surfaces."*

to:

*"Assuming the lowest albedo of 0.04, and a panel efficiency of 0.1-0.15, this would yield effective albedo ($A_{eff}$) values of 0.14 and 0.19, i.e., similar to jojoba and jatropha. Although solar PV panels may be a suitable subject for future research into rainfall enhancement, this study focuses on simulation of generic black-painted panels with a set of prescribed parameters to describe the likely land surface properties of such a surface. In this study we use the term 'Artificial Black Surfaces (ABS)' for these panels."*

**To simulate this, would have to modify the surface energy budget to have a special "energy out" term that is energy production; this would be the most physically consistent way. Instead, it sounds like the authors have imposed surfaces brighter than actaul PV panels, such that the energy that typically would go to energy production is instead reflected away from the surface as SW radiation. For the most part, this probably isn't going to qualtiatively change their answers given the magnitude of changes imposed here. But the atmosphere \*does\* absorb SW radiation, including SW reflected from the surface, and also altering SW albedo in these low water vapor, relatively cloud-fre desert regions will alter the top of atmosphere energy balance in a way that could influence circulation.**

Thank you for these well-considered ideas. Given that we have now narrowed the focus toward black-painted panels we consider it is therefore no longer necessary to delve too deeply into PV properties. However, the radiation exchanges you discussed are very interesting and we are considering future publications on solar PV including both measurements and simulation components.

**If you instead had the surfaces as dark as they actually are, with an extra term in the surface energy budget that removes energy used for power generation, you'd have to release that energy \*somewhere\* in a coupled model to conserve energy. In a regional model, though, you could just assume that the energy is moved out of the region you're simulating. I'm not suggesting you re-do your simulations this way - just that you explain what is and is not representative of the actual physical system (real world) in the way youv'e chosen to simulate this.**

Please refer to comment above.

**And of course, if the surface is made artificially dark without any power production - e.g. just painting surfaces dark, which the authors do discuss as one option of ABS - the power / energy conservation thing isn't a concern!**

Please refer to comment above.

**Line 56: What is reducing the cell efficiency with precip? Being low efficiency makes it lower efficiency with time? Or getting rained on makes it lower efficiency with time?**

Please refer to comment above. We have removed this reference to the efficiency reduction now, along with the citation (L56-57):

*"This may be useful for rainfall enhancement, but can reduce cell efficiency and longevity (Dwivedi et al., 2020)."*

The authors hope that the manuscript is more clearly defined now.

**Line 84: Is "material" really appropriate here? Maybe "Model and Methods"?**

Agreed. We have changed the section title as you suggest to "Modelling and Methods".

**Line 87: specify the region (title just says "arid regions", and intro discussed the Middle Eastern Gulf region, but here specify that that is indeed where you're going to simulate, and maybe modify the title to reflect the actual region of study – it would be a leap to extrapolate from this analysis to all arid regions, as the background flow and moisture sources are pretty critical to the results).**

Thank you for this important point. We suggest modifying the title to "Scaling Artificial Heat Islands to Enhance Precipitation in the United Arab Emirates".

To specify the region, we have also amended the text (L87) from:

*"Simulations were carried out with the Weather Research and Forecasting (WRF) model (V4.2.1, Powers et al., 2017). WRF has been used in the region for numerous model evaluation and process (Branch et al., 2021; Fonseca et al., 2020; Valappil et al., 2020; Wehbe et al., 2019; Nelli et al., 2020; Schwitalla et al., 2020), and rainfall modification studies (Mostamandi et al., 2022; Wulfmeyer et al., 2014; Branch & Wulfmeyer, 2019)."*

*to:*

*"Simulations were carried out with the Weather Research and Forecasting (WRF) model (V4.2.1, Powers et al., 2017). WRF has been used **in the middle-east region** for numerous studies on model evaluation and processes (Branch et al., 2021; Fonseca et al., 2020; Valappil et al., 2020; Wehbe et al., 2019; Nelli et al., 2020; Schwitalla et al., 2020), and rainfall modification (Mostamandi et al., 2022; Wulfmeyer et al., 2014; Branch & Wulfmeyer, 2019)."*

**Line 90: in the same domain? Please specify.**

Thank you. To clarify that this is a model domain, we have added the word "model" in the text (L90) to:

*"Here, we use the same **model** domain, resolution and configuration"*

**Line 92: MYNN - what is this?**

Thank you for alerting us to this oversight. This refers to the Mellor-Yamada-Nakanishi-Niino boundary layer scheme, selectable in WRF (Nakanishi & Niino, 2006). We will clarify this by modifying the text (L93):

*"...to take advantage of improvements to the **Mellor-Yamada-Nakanishi-Niino (MYNN)** boundary layer,..."*

**Line 98: I know the choice of convection scheme needs to be specific to the model resolution, but am not up-to-speed enough with the WRF convective scheme options to know what the appropriate choice here is. Also, please specify which scheme you used.**

This is a very good point and may be clarified. As the model is being run at the so-called 'convection-permitting' scale (usually ~<4 km) convection is not parameterized but is simulated explicitly. We clarify this by adding the following line (L94):

*"The model is being run at a 2.7 km grid increment, which lies in the so-called 'convection-permitting' (CP) scale (generally <4 km), which allows that convection can be simulated explicitly, and not parameterized."*

We hope this is now clear.

**Line 105: are you still talking about the outside of domain forcing? Clarify. Sounds like you're talking about the model, but the land surface you use in the model domain is NOAH-MP, right?**

This does indeed relate only to the global IFS model proving the boundary and initial conditions, which has its own land use model, HTESSEL. Within the WRF domain, the land surface processes are carried out by the NOAH-MP scheme. The only exception to this are OSTIA sea surface temperatures which is the only data re-initialized periodically *within* the WRF domain itself.

We clarify this by modifying the text (L107-108) from:

*"Additionally, OSTIA sea surface temperature (SST) data ($\Delta x$ 1/20°, Donlon et al., 2012) were also ingested every 12 hours (00:00 and 12:00 UTC), which is particularly important for simulating sea breezes."*

*to:*

*"The ECMWF forcing data is only used to provide the lateral boundary and initial conditions for WRF-Noah-MP which then itself develops the evolving conditions within the model domain. The only exception to this, is the ingestion of OSTIA sea surface temperature (SST) data ($\Delta x$ 1/20°, Donlon et al., 2012), which are re-updated within the domain every 12 hours (00:00 and 12:00 UTC). This is likely to be beneficial for simulating the sea breeze."*

**Figure 1: what is the 899x699x100? number of x, y, and z grid cells?**

We agree this could be expressed more clearly. We have amended the line (L101) to:

*"The model grid has horizontal dimensions of 899 (east-west) × 699 (north-south) cells, and 100 vertical levels."*

**Figure 1: Could you please make ocean a different colour than low (0-400m) land?**

Thank you. The figure has now been modified accordingly.

[Figure]

**Line 128-131: this discussion is a great orientation to the base-state of the region! nice!**

Thank you for this positive comment.

**Line 136: what four selected cases? To help the reader not get confused, it would be helpful if before this point, you say that you're going to do, for each ABS box, 4 different runs of YYYY days each, each forced with weather conditions from **

This is a good suggestion. To better summarize the modelling method, we have accordingly modified the line (L137) from:

*"We illustrate weather conditions for the four selected cases in Figures 2 and 3 (from Control) to highlight the likelihood of impacts."*

to:

*"In this study we selected four one-day case studies. For each of these days, we simulated the five ABS scenarios (10, 20, 30, 40, 50 km squares) and a Control simulation for a total period of 24 hours (00:00-00:00 UTC). We illustrated the respective weather conditions for our four selected cases in Figures 2 and 3 (from Control) to highlight the likelihood of impacts."*

**Line 137: "Typical CI…" --- Define (I assume "convective initiation")**

This was already introduced. However, to avoid overuse of acronyms we have changed 'CI' to 'convection initiation (CI)' at this location.

**Figure 2: Adding a map, either as a separate figure or in one of figure 1 or 2, that shows the prevailing near-surface and aloft wind directions (on the map) for these two time slices would be really helpful. (Figure 3 has this for 10am, I assume near the surface? But maybe not near the surface – not specified!)**

The wind field shown in Figure 3 is 10-m winds at 10:00 UTC. This is marked both in the caption and the text. Prevailing winds aloft are shown as wind barb profiles in the Skew-T plots of Figure 2 at both 06:00 and 10:00 UTC. We are confident that together, these provide a sufficient picture of the wind field at the relevant locations and times (pre-sea-breeze and at the time of the seabreeze).

**Figure 3: what are 18.07, 24.07, 25.07, and 27.07? Can you write "July 18", "July 24", etc instead? otherwise this reads like a number and the reader gets confused about what the number means (when in fact it is a date)**

Agreed. The date format throughout text and figures has now been modified to e.g. 'July 18'.

**Figure 3: figure 3 b, c, d - make the reference vector larger, like in panel a --- otherwise it is too tiny for my poor eyes...**

Agreed. The larger reference vector has been added to the other panels.

[Figure]

**Figure 3: what level are these winds from? surface?**

This is stated in the Figure 3 caption:

*Figure 1: Control thermodynamic conditions at 10:00 am (UTC) during the four case studies, with red boxes to show the relative position of the ABS. The top row shows convective available potential energy (CAPE, J kg⁻¹), **10-m wind vectors**, and a box showing the vector reference length for 3 m s⁻¹ (panel (a)). The bottom row shows convective inhibition (CIN, J kg⁻¹). These conditions were used to investigate the daily sea breeze timing and select our case studies.*

And also in the text at L142:

*"Figure 3 shows a horizontal perspective of convective available potential energy (CAPE), convective inhibition (CIN), and **10-m winds** (at 10:00 UTC)."*

**Figure 4: Again, please label with "July 18, July 24" etc. What are the arrows for? Wind vectors on the control would be helpful, but I don't think that is what the arrows are for here?**

We agree, and have modified the date format throughout. The arrows were just to indicate that the panels below are a modification of the Control. However, as the arrows are perhaps more confusing than useful, we have now removed them. Regarding the wind vectors, the authors have considered this comment. Firstly, we think that because the figure shows precipitation totals of a whole day, it is probably not so useful to see wind vectors from a certain timestep, or even a daily vector average (which given the complex changes in daily wind flow, could be misleading anyway). Finally, we think that in such small panels, wind vectors would make the plot far too busy in any case.

[Figure]

**Figure 4: Inconsistency between text (line 180) and figure caption. Do you calculate the precipitation response in a circle of radius 90 km (ie diameter 180 km), as on line 180, or a circle of diameter 150 km (radius 75 km) as in the caption of figure 4?**

Thank you for noticing this error! The caption is incorrect and should read 180 km diameter. This is now changed.

**Line 200: panel e of what? Sounds like you're talking about figure 5 in this paragraph, but figure 5 only has panels a-d.**

Thank you for pointing out this error. It is panel b. We have modified the text accordingly (L201) to *"(Figure 5, panel b)*

**Figure 5: clarify "millions of m3" somewhere**

Thank you. We have now added "*The amounts are in million $m^3$*" To the Figure 5 caption.

**Figure 5: again, please write "July 18, July 24..."**

Agreed. This now done.

**Line 207: clarify this is combined accumulated precipitation volume for \*both\* regions**

Thank you. This is already indicated in the text (L198) with "*(sum of east and west circles)*". We have now added this same text to the caption.

**Line 207: "…*UAE per capita supply based on 500-liter capita-1 day-1 (right axis, people yr-1), amongst the*** ***highest in the world*" --- I read this as the UAE having the highest per capita supply of water in the world... is that right? That is not intuitive to me! Or is it the volume of supple that is high? Please clarify.**

The UAE does indeed have one of the largest per capita water consumption rates in the world. To clarify this and emphasize the importance of protecting water resources in this region, we have modified the text (L207-209) from:

*"Panel (b) shows the difference between the ABS and Control expressed in volume (left axis, mil. $m^3$) and UAE per capita supply based on 500-liter capita$^{-1}$ day$^{-1}$ (right axis, people yr$^{-1}$), amongst the highest in the world (Albannay et al., 2021; Yagoub et al., 2019)."*

to:

*"Panel (b) shows the difference between the ABS and Control expressed in volume (left axis, mil. $m^3$) and UAE per capita supply based on 500-liter capita$^{-1}$ day$^{-1}$ (right axis, people yr$^{-1}$), which is one of the highest national consumption rates in the world (Albannay et al., 2021; Yagoub et al., 2019)."*

**Line 212: "*There is a small surplus on 24 July though.*" --- Just for the 50 km case, right? No, I'm confused - what do you mean "surplus"? I don't see the demand listed anywhere on thesee plots, so what does a "surplus" mean?**

The authors agree with your comment that 'surplus' may perhaps not be the best word to use here. We have therefore changed the word 'surplus' to 'increase over Control' at (L189 and L213).

**Figure 6: So, Panel C is showing that the 50 km blobs of ABS lead to almost a 50% increase in precipitation, is that right? That seems worth highlighting!!!**

The authors agree with your assessment of this significance. At your suggestion we decided to add some text to emphasize this. We have added a short sentence here (L227):

*"The mean increase from both 50 km ABS represents almost a doubling of the rainfall in Control."*

**Line 258: influence of what from the surrounding what?**

Here the authors are referring to the influence of the surrounding physical environment on the air mass inside the ABS zones. This could come from advection, entrainment or other processes. To clarify this, we have modified this sentence (L259) to:

*"e.g., due to influence from the environment surrounding the ABS, such as from advection or entrainment of heat, moisture, or other quantities"*

We hope this has improved clarity here.

**Figure 7: it would be helpful to highlight the zero line here, e.g. with a darker horizonal line - at least in the pdf I have, there is no / almost no line visible at 0, 50, and 300. The one at 0 is particularly important to have though, and to make stand out more than the rest!**

Agreed. The png format uploaded with the submission are lower resolution than the final tiffs that will be provided. Hence the faded lines. We have now added a thick reference line at 0, and show the final high resolution tiff image here.

[Figure]

**Text/Grammatical:**

**Line 30: "are become" - "are becoming" or "have become"**

Thank you. This line now reads: *"Regional crises like high temperatures, drought, wildfires, flooding and water scarcity are becoming more severe"*

**Line 258: "flu" -> "flux"**

Thank you. Corrected.

**Concluding remarks from the authors:**

Many thanks again to the Reviewer for taking the time to review our work and for providing well-thought out and constructive comments. We feel that addressing your comments has greatly improved the manuscript.

---

## Author Comment (AC2)

**Point by point reply to Reviewer 2**

**Review for "Scaling Artificial Heat Islands to Enhance Precipitation in Arid Regions"**

**Opening Reviewer comment.**
**The authors present a study of the impact of artificial reductions in surface albedo as a means to enhance convective precipitation over the hyper-arid UAE in a convection-permitting modeling framework. These reductions in surface albedo are imposed as uniform changes to the land surface albedo over a prescribed area in the model, but in reality would be achieved through some combination of solar panels, vegetation plantations, and other artificial surfaces that are much darker than the surrounding desert. Fundamentally, these changes in albedo impact precipitation in the model through their enhancements of surface sensible heat flux, which in turn has a variety of meteorological consequences. The sensitivity of precipitation impacts to an increasing area of albedo modification is also a key consideration.**
**I found the study to be reasonably well constructed and executed. The modeling tools and analysis seem appropriate for the key questions of the work. It's an interesting and relevant topic of study, and I am curious how this work will continue or even be applied in the future.**

We would like to thank the reviewer for taking the time and effort to review this submission, and for the positive appraisal which is much appreciated. We would like to address your comments below, point by point:

**Comments**
**I believe the paper could be improved through an expansion of the context provided first in the Background section, which the authors then return to in the Summary and Outlook. For example,**
**● Line 33: The authors reference a few papers presumably showing that albedo "can trigger regional scale impacts," but they don't provide any detail on what those previous papers were studying and whether they are relevant to this paper in particular. If you are just trying to generally say that albedo can alter regional climate and weather, I'd also look for a bit more detail here highlighting some of the ways that people have shown albedo changes altering regional climate and weather. But it would be especially good if they were directly relevant to this work and connected into the introduction more smoothly.**

The authors agree that perhaps a bit more background could be useful here regarding albedo change. These papers (Doughty et al., 2011; Kala et al., 2022; Ridgwell et al., 2009) are thought to be relevant here because they relate to *deliberate* albedo change. However, these all relate to an increase in albedo, not a local decrease as we are proposing here. Of course, increasing albedo on a wider scale is also a contribution to a reduction in overall global forcing (however small..) as well as a potential means of modifying regional climate. Papers which advocate deliberate albedo *reduction* are scarce, aside from four of our own studies which all relate to this topic and together provided the basis for this work (see also the next point).

To address your important point, we have modified some text to emphasize that the three studies (Doughty et al., 2011; Kala et al., 2022; Ridgwell et al., 2009) relate to deliberate albedo increase, or 'brightening', and to clarify the methods used (L31-35).

From:

*"Examples are marine cloud seeding to reduce coral bleaching (Latham et al., 2014; Latham et al., 2013; Tollefson, 2021) and albedo management through landscape planning, and breeding of higher-albedo crops (Doughty et al., 2011; Kala et al., 2022; Ridgwell et al., 2009). Albedo change can trigger regional scale impacts (Quaas et al., 2016; Seneviratne et al., 2018) such as reduction of temperatures (Kala & Hirsch, 2020), but at the same time could also contribute toward reduced global forcing (Carrer et al., 2018; Sieber et al., 2022)."*

*To:*

*"Examples are marine cloud seeding to reduce coral bleaching (Latham et al., 2014; Latham et al., 2013; Tollefson, 2021) and **deliberate** albedo management through agricultural landscape planning, and breeding of higher-albedo crops (Doughty et al., 2011; Kala et al., 2022; Ridgwell et al., 2009). Their general aim is to investigate the potential for regional cooling of temperatures. **The deliberate increase of albedo falls under the**

*geoengineering category of terrestrial solar radiation management (SRM). And although geoengineering is considered to be global in scale, regional actions may trigger regional impacts (Quaas et al., 2016; Seneviratne et al., 2018) such as reduction of temperatures (Kala & Hirsch, 2020), whilst at the same time contributing toward reduced global forcing (Carrer et al., 2018; Sieber et al., 2022).*

**● Lines 40 - 45: Branch and Wulfmeyer (2019) seems very relevant to your work here, and I would like to see a bit more drawn from that paper into your introduction in terms of what they found, how your approaches are similar/different, how it might have motivated this study etc…**

Absolutely. Our previous works on deliberate albedo change are directly related to this work (Branch and Wulfmeyer, 2019; Wulfmeyer et al., 2014; Branch et al., 2014; Becker et al., 2013). These works all have a different emphasis though and relate to deliberate local reduction in albedo through cultivation of desert *vegetation*. Our work differs in that here we focus on artificial non-vegetation surfaces, and also that we examine the effects of scale and associated lower boundaries for impacts, which the other works did not.

To address your comments and clarify these links between our previous and present studies, we have re-emphasized that these works were carried out by our group, which is important to set our latest work in context and build on the findings and new process understanding. Therefore, we have modified the text at L39-40:

From:

*"Several studies show that desert xerophyte plantations can enhance rainfall via canopy heating (Becker et al., 2013; Branch et al., 2014; Branch & Wulfmeyer, 2019; Wulfmeyer et al., 2014)."*

*To:*

*"Several of our previous studies show that desert xerophyte plantations can enhance rainfall via canopy heating (Becker et al., 2013; Branch et al., 2014; Branch & Wulfmeyer, 2019; Wulfmeyer et al., 2014)."*

To link our most relevant publication (Branch & Wulfmeyer, 2019, PNAS) to this one we have also modified the text at the end of the introduction (L82-84).

*From:*

*"In the Results and Discussion section, we present impacts on precipitation, convective processes, and feedbacks. Finally, in the Summary and Outlook section we put results in context and discuss wider implications."*

To:

*"In the Results and Discussion section, we present impacts on precipitation, convective processes, and how we applied a thermodynamic feedback index to investigate predictive potential (similarly to Branch & Wulfmeyer, 2019). Finally, in the Summary and Outlook section we put results in context and discuss wider implications."*

The authors hope this addresses your comments without increasing the length of the paper too much.

**I also believe the paper could benefit from more detail on prior work looking at convection & precipitation and its diurnal cycle in the region more broadly, at least during the summertime months considered. How often does it occur? Does it go right to deep precipitating convection or is there shallow convection first? What's the variability? Are there any consistent patterns, and when might those patterns break down? I think this would feed well into your proposed future work directions at the end of the paper, expanding into other times of year or if there's climate variability. It would just be helpful to know more about what the background setting of convective precip is in the region you're looking at, in case the reader is less familiar. In other words, what exactly are we modifying through the albedo perturbation in the first place!**

This is a very good idea. We did provide some background already on the climate and precipitation patterns in the region, and have cited our previous work on convection initiation (CI) patterns in the area (Branch et al, 2020) on identification of CI events from Meteosat radiances. Perhaps we did not reference this work strongly enough. This paper shows that deep convection only occurs with any regularity in the eastern Al Hajar Mountains, and areas west of these mountains are in general extremely arid, dominated by regional subsidence. Given that our ABS patches were simulated on these plains we have emphasized this more within the text and suggested to the reader to investigate our work on convection patterns in the region (after L131-132). We have also added a line to state the main climatic 'constraint' to be overcome by the land surface modification method:

Lines modified from:

*"Warm sea breezes reach the coast around midday (local time (LT), Eager et al., 2008), and reach the ABS areas typically around 14:00 LT."*

To:

*"Warm sea breezes reach the coast around midday (local time (LT), Eager et al., 2008), and reach the ABS areas typically around 14:00 LT. Nevertheless, deep convection only occurs regularly in the eastern Al Hajar Mountains. The more westerly desert plains are in general extremely arid, dominated by regional subsidence and capping inversions (See Branch et al. (2020) for regional convection initiation statistics). These inversions must be overcome by any deliberate albedo modification"*

We trust that this addition provides enough detail and has improved the context.

**Similarly, I think the authors need to provide a bit more detail on how the model they are using does with convection and precipitation in this region. They note a validation study in Line 90 from a few years ago, but they don't indicate whether the results of that evaluation were favorable, especially for the variables they're pulling out of WRF in this analysis. Does the model reproduce CAPE over the UAE well compared to soundings, for example? Could you show satellite imagery, if radar isn't available, showing that your control simulation produced reasonable patterns convection on your four case days?**
**This is also important to discuss in your Model Configuration section if the only prior validation was done against surface observations. And along those lines – I would be careful in saying that this version of WRF is the same as the "validated" version when the "only change" was to use an updated version of the PBL, surface layer, and land schemes (Line 92) – the components of the model that are among the most important for this study. Just because a model component has been updated doesn't guarantee that it will improve the skill, especially over a particular region and when looking at something as sensitive as convective precip. Overall, I would just be more clear about what has been and what hasn't been validated for this particular region using this configuration of WRF. And it may be the case that some aspects (like my CAPE example above) haven't been tested exactly, but I would just mention that as a caveat and/or an area for future work.**

The authors agree that verification is very important. Our previous validation of the WRF model (Branch et al. 2021) was based on a computationally-ambitious one-year model simulation, evaluated with the Model Evaluation Tools (MET) package against data from 48 surface stations in the UAE. We are convinced that this provided good context as to the skill of WRF regarding simulation of regional surface conditions. We will include some details on the evaluation results here as you suggest, particularly as daytime temperatures and dewpoint were satisfactorily reproduced during daytime, for the most part. It is for this reason that we used the same resolution, configuration, and forcing/sea surface temperature data for this study (although an updated WRF version). The only persistent deficiency in the model was an overestimation of daytime surface winds, which was observed also in some other papers (Fonseca et al., 2020; Temimi et al. 2020) [In an oversight, this last Temimi 2020 paper was not cited and so we have added this to the manuscript (L88)].

The successful simulation of convective precipitation is always a challenge for numerical models. The expectation that case-study downscaling simulations can reproduce actual cloud and precipitation patterns at any point in time is likely to be unreasonable *without* a significant data assimilation effort, which is beyond the scope of this impact study. For these reasons, we do not evaluate these phenomena for our cases, but instead rely on findings from other

regional studies. However, we have now expanded the text in the Summary and Outlook to further emphasize the need for more testing (L458):

*"Ensemble UAE simulations with varying model physics including microphysics (e.g., Schwitalla et al., 2020; Fonseca et al., 2020, Taraphdar et al., 2021) would be particularly useful. Here, data assimilation, quantitative precipitation estimation, and rain radar and satellite analyses may be employed (Bauer et al., 2015; Kawabata et al., 2018; Branch et al., 2020; Schwitalla & Wulfmeyer, 2014).* **The latter analyses, probably based on seasonal simulations, will be especially important to test further the plausibility of convective rainfall amounts, from a statistical point of view.** *For now, we have at least some confidence that WRF V4 has reproduced convective cells in a satisfactory way in the region (Fonseca et al., 2022."*

As well as the studies that we have already cited around L88-89, we have now added (in Modelling and Methods and in the Summary and Outlook) a more recent study by our UAE colleagues who recently assessed the usefulness of WRF V4 in predicting cloud and convective patterns, with found satisfactory results (Fonseca et al. 2022). We trust that this adds weight to confidence in the model performance in simulating convective process (especially with WRF V4 which we used here).

To address your comments, we added more detail to the verification results and also added/modified text (L90-92) from:

*"Here, we use the same domain, resolution and configuration to Branch et al., 2021, who evaluated the skill of WRF (V3.8.1) in reproducing 2-m temperature and humidity when compared to 48 surface weather stations."*

*To:*

*"Here, we use the same domain, resolution and configuration to Branch et al., 2021, who evaluated the reproduction of 2-m temperature and humidity by WRF (V3.8.1) in when compared to 48 surface weather stations. The results over four seasons were satisfactory, providing a good basis for using this model at the same scale. Here we used an updated version of WRF (V4.2.1), and make the assumption that the model performance has not deteriorated with the model updates. Indeed, Fonseca et al. (2022) assessed the ability of WRF V4 to reproduce cloud and convective cell spatial distributions, for the purpose of cloud seeding operations, and with satisfactory results. This adds some confidence in the reproduction of convective process in the UAE by WRF V4, as used within this study."*

We have also exchanged the word 'improvements' at L93 to 'updates' to avoid the impression of unwarranted claims about performance.

**Additionally, I think you have an opportunity to better connect your work to other areas of surface-atmosphere interactions in the Background (Lines 60 - 72) and the Outlook to round out the manuscript. Enhancements in sensible heating, usually associated with changes in vegetation cover or properties, have been shown to alter cloudiness/convection/precip in different ways depending on where you look. Your work highlights the importance of background humidity (here brought by the daily sea breeze) that can then be lofted to saturation by deeper, more vigorous boundary layers, which is something that also comes up over vegetated surfaces. I think highlighting that similarity and connection, especially in an environment with little-to-no latent heat flux would ground the study in prior/ongoing work more completely.**

This is a good suggestion. We are very conscious about keeping the manuscript length relatively concise, and not including too much general background on land atmosphere interactions over multiple scales. We elected to maintain the focus only on this particular effect, i.e., a relatively localized isolated heat perturbation, especially in arid environments. Hence, we elected to include only the most relevant papers, relating to this effect and at similar meso-gamma to meso-beta scales e.g. urban heat islands. To address your comments though, and to introduce a link to moisture requirements in arid regions (with low latent heat fluxes), we have modified the following sentence (L39-40) from:

*"Several studies show that desert xerophyte plantations can enhance rainfall via canopy heating (Becker et al., 2013; Branch et al., 2014; Branch & Wulfmeyer, 2019; Wulfmeyer et al., 2014). Branch et al., 2014, measured albedos of 0.17 and 0.12 for jatropha and jojoba plants, and the surrounding desert ~0.3, leading to temperatures*

*up to 4°C higher than the surrounding desert (see also Saaroni et al., 2004). This heating led to greater simulated cloud development and ls (CI) (Branch & Wulfmeyer, 2019)."*

*To:*

*"Several of our previous studies show that desert xerophyte plantations can enhance rainfall via canopy heating (Becker et al., 2013; Branch et al., 2014; Branch & Wulfmeyer, 2019; Wulfmeyer et al., 2014), facilitated by the advection of coastal marine moisture. Branch et al., 2014, measured albedos of 0.17 and 0.12 for jatropha and jojoba plants, and the surrounding desert ~0.3, leading to temperatures up to 4°C higher than the surrounding desert (see also Saaroni et al., 2004). A subsequent model simulation with a similar magnitude of heating showed increased cloud development and convection initiation (CI) (Branch & Wulfmeyer, 2019). "*

**My final "general comment" deals with the temperature effect of the albedo perturbation introduced by the ABS. This study is mainly looking at precipitation effects, which makes sense given the region, but I would also look for some discussion of other implications of this strategy. If we darken the surface, we will also increase near-surface air temperatures. Will this be a problem locally for people, even if it is helping with some of their water scarcity issues? Maybe the temperature change isn't impactful relative to the background hot climate.**

Both Reviewers raised this very important point, and from the beginning we did consider the potential impacts on regional temperatures. We suggest adding a new figure, if space allows, showing the mean impact on simulated daily maximum and mean 2-m temperatures:

[Figure]

*Figure X: The case-average impact on daily mean and daily maximum 2-m air temperatures from the 50km ABS. Computed respectively, as $\frac{1}{n}\sum_1^n \left(\overline{T2m}\ ABS_{50km} - \overline{T2m}\ Control\right)$ and $\frac{1}{n}\sum_1^n \left(T2m_{max}\ ABS_{50km} - T2m_{max}\ Control\right)$, where n is the number of cases. Panel (a) is the daily maximum 2-m temperature impact, and Panel (b) 24-hour daily mean impact.*

To supplement the figure, we would add more text (at L241) to include information on how surface heating is distributed both inside and around the ABS zones. We have added the following text:

*"It is also important to consider the impacts on temperatures outside as well as inside the ABS zones, because large temperature changes could affect local citizens and vegetation. Figure X shows the difference (case-average) in daily maximum and mean temperatures between the 50 km ABS scenario and the Control. For maximum daytime temperatures (panel a), there is a maximum temperature increase of up to ~1 °K, inside the 50km ABS zones when compared to Control. In the surrounding areas, there is a temperature increase particularly around the eastern zone, but the differences are relatively limited, both in spatial extent and the temperature increase (~0.2-0.3 °K). Curiously, there are also some minor cooling effects to the south of the ABS. For daily mean temperatures (panel b), there is an increase of up to ~0.8 °K inside the ABS zones. Outside the ABS, the largest increases are ~0.2-0.3 °K, but these areas are quite close to the ABS zones. These simulated values indicate that there is a slight temperature impact on the near-surroundings, but even at the largest ABS 50 km scale, this is simulated as low to moderate."*

**If the area of the albedo perturbation gets large enough or there are too many of them, could it alter sea breeze or other regional dynamics in unhelpful (or maybe helpful) ways? If this is being considered in the context of a regional climate strategy those other effects need to be mentioned, at least. And again, that might be an area of future work for this manuscript, but I do think it needs to be mentioned.**

The authors agree. We have added an additional line in the Summary and Outlook section to state that there are wider uncertainties on climate impacts which may need to be addressed in future works (L432):

*"It is still uncertain though if impacts on circulation or the sea breeze are modified, or even tele-connected to other regions. This is subject requiring further investigation, most likely with longer simulations to provide statistics."*

**Minor/Detailed                                                                           Comments:**
**● Line 47: what kinds of weather modification have the previous studies shown for the PV panels? This ties back into my general comment above about more detail in the background, but just wanted to make a specific note here.**

Studies on deliberate use of solar photovoltaic for weather modification are scarce. We have cited one study (Mostamandi et al., 2022) who state that simulated solar panels over the Saudi Arabian coast increases surface air temperature by 1–2 K, strengthens land-sea temperature contrasts, intensifies breezes, increases water vapor mixing ratio in the boundary layer, and increases rainfall. However, partly based on other Reviewer comments, the authors have elected to reduce emphasis on the use and description of solar panels somewhat, and we instead focus on black panels for the ABS systems. The use of solar is still given as a potential surface given the proliferation of solar farms in the region but we have reduced the references to solar PV in the Background section. One good reason for this is that, in our opinion, a good representation/simulation of solar surfaces (with varying cell materials), requires detailed solar farm measurements – which may form part of a future study.

**Also, I don't believe you ever spelled out PV as photovoltaic, which would be good for the first use.**

Thank you. This has now been amended.

**● Line 57: somewhat tangential, but are those PV panel efficiencies valid for an environment like the UAE with such high temperatures?**

This is a good question, but we had anyway considered possible efficiencies as low as 10% in the text, which we think is reasonable. Nevertheless, as we mentioned above, we have reduced our emphasis on solar PV.

**● Line 70: found "interactions" between urban heat islands and sea breezes, particularly for convection – what are those interactions? Are they relevant for your results here?**

This is a very good idea. We have added here some text on the interaction between sea breeze and urban heat islands (L72):

*"Some studies include the interactions between UHIs and sea breezes (e.g., Cenedese & Monti, 2003; Freitas et al., 2007; Zhang & Wang, 2021), and found interactions between them, particularly for convection. For instance, Freitas et al. (2007) found that an urban heat island (UHI) formed convergence zones in a city, accelerating the sea-breeze front toward the city centre."*

**● Line 83: The term Artificial Heat Island is not used in the introduction. Given how it's used to frame the study in the paper title, I would look for it to show up here somewhere. Perhaps in the connection from Urban Heat Island.**

An excellent idea. We have now modified the sentence at L72 to:

*"The relevance of this interaction will become apparent later in this study on 'artificial heat islands'."*

● **Line 105: Can you clarify that this data assimilation of soil moisture and temperature is happening in the ECMWF system you use for your boundary conditions? Or is this going into WRF directly?**

This was a question also raised by another reviewer and we have clarified this at L107-108:

*"The ECMWF forcing data is only used to provide the lateral boundary and initial conditions for WRF-Noah-MP which then itself computes the evolving conditions within the model domain. The only exception to this, is the ingestion of OSTIA sea surface temperature (SST) data (Δx 1/20°, Donlon et al., 2012), which are re-updated within the domain every 12 hours (00:00 and 12:00 UTC). This is likely to be beneficial for simulating the sea breeze."*

● **Line 136: Could you provide more detail on how the case days were selected within the year and if the days themselves are representative of diurnal patterns in the study area (I see that you noted a validation that summer 2015 was climatologically representative but not the days themselves)?**

In fact, we are looking for a representative season, and year, but we are not looking for 'representative' days for our cases, but days on which convection is likely with some heating perturbation, e.g., moderate to high CAPE and moderate CIN. We outlined our criteria here (L133-134):

*"Our aim was to observe strong impacts, so we selected a season where impacts are most likely (summer) and days with unstable convective conditions during a suitable season.We elected to do the simulations in JJA 2015, to coincide with the evaluation simulations of Branch et al. (2021), who found this season was representative in terms of the long-term climate. Summertime was also selected because strong impacts were observed during this season (e.g. Branch and Wulfmeyer, 2019)."*

Also, in L167-168, we have also stated:

*"In summary, all cases are moderately unstable and exhibit a wave of reduced CIN passing over the ABS. CIN may be a defining constraint, for even when CAPE is only moderate (e.g., 27 July), a low CIN may still permit CI."*

The authors trust that this provides enough justification on our year/season/case selection.

● **Line 154: When you say conditions in the east vary more with the Gulf of Oman, can you be more specific?**

Agreed. To some extent we did specify this in the second sentence here:

"In the east, conditions vary more with influence from the Gulf of Oman. Sometimes wind confluence occurs (e.g., 27 July, Figure 3, panel d), or one side dominates the other, or more southerly winds prevail."

But perhaps the link between the two sentences is not clear. Hence, we have modifed this now to:

*"In the east, conditions vary more with influence from the Gulf of Oman, with wind confluence from these two Gulfs occurring (e.g., 27 July, Figure 3, panel d), or sometimes the winds from one direction dominate the other, or at other times more southerly winds prevail."*

● **Line 177: Is the model overly drizzly?**

The model is not particularly drizzly in this simulation. Nevertheless, we have maintained the lower (Frey) threshold of 1mm to remain consistent with current rainfall modelling studies.

● **Caption for Figure 4 references a 150 km diameter, but in the text it is 90 km.**

Thank you. This was a caption error and has now been changed to 180 km diameter.

● **Lines 230 - 240: How confident are you in the ability to operationally predict where precip enhancements needed to be captured for human use? Is the enhancement based on the ABS here falling in the right spot that it could be collected or directed to groundwater recharge? Would be important to note if this is being used to justify any sort of deployment/construction.**

This is a very interesting point. The practical implementation of water collection is not part of this study, and we simply assume that all extra precipitation is a 'benefit' whether it is collected or put in the groundwater. A more crosscutting his would certainly be an interesting investigation for later publications.

● **Line 245: I'm curious why you didn't pick the case where the rainfall impact was strongest?**

We elected to choose not the most extreme case, but instead a moderate one from our four cases. We feel it illustrates the convective processes very well.

● **Line 258: "heat flu" is missing the x for "flux"**

Thank you. Corrected.

● **The caption on Figure 7 was a bit confusing in terms of the left panel having "both 50 km ABSs", when you're just noting that they have a common footprint**

We agree that this could be expressed more clearly. Hence, we have re-written the caption as:

*"Figure 1: Mean sensible heat flux and standard deviation on 27 July. The left panel shows the spatial mean within a common footprint (the 50 km ABS area) for all scenarios The right panel shows the spatial mean of each individual ABS footprint. Here, two spatial means are shown for Control - the 10km and 50 km footprints, so they can be compared."*

We trust that this provides more clarity for the reader.

● **Line 283: Why were these other factors disregarded?**

This was clarified in the following sentence but could be clearer. Hence, to connect this reasoning, we have modified this sentence from (L283-285):

*"Differences in static land surface characteristics, such as soil texture or moisture, were considered as possible reasons for these patterns but these were disregarded. The soil moisture is virtually zero and the soil texture is very homogenous over the whole area."*

*to:*

*"Differences in static land surface characteristics, such as soil texture or moisture, were considered as possible reasons for these patterns but these were disregarded, because the soil moisture is virtually zero and the soil texture is very homogenous over the whole area."*

● **Line 324: "diurnal timing" of what?**

Agreed this needs clarification. We have modified the sentence from:

"Interestingly in the 50 km scenario, low-level convergence and PBL development are impacted earlier in the day, indicating that diurnal timing may also be important for CI."

To:

"Interestingly in the 50 km scenario, low-level convergence and PBL development are impacted earlier in the day, indicating that the varying diurnal onset of strong convergence between the scales may also be important for triggering CI."

● **Line 352: CI referring to convective initiation or impacts?**

This refers to convection initiation here. We have added the words "convection initiation is influenced" here (L353-354):

Our goal is to gain insights into how scale influences convective initiation, and it is known that convection initiation is influenced not only by large-scale conditions, but also by land-atmosphere (LA) interactions, or feedbacks (e.g. Jach et al., 2020, Gerken et al., 2019).

● **Line 426: Just to clarify, is the HCF index what you are using as your LA feedback metric?**

Yes this is correct. To make that clear we have modified the sentence (L427):

*"and to explore the predictability of impacts by applying the LA feedback metric, the HCF index."*

*To:*

*"and to explore the predictability of impacts by applying the LA feedback metric (the HCF index)."*

● **Line 453: Are the water quantities produced by the model microphysics plausible? This could tie back into my question about more detail about model validation/future work and the context of convective precip over the reg**

This is of course an excellent question, and we have written in the text that we are assuming that precipitation amounts are *reasonable* at L454:

"In terms of rainfall enhancement, **if we assume** that the water quantities produced by the model microphysics are plausible, then the implications are considerable."

We will make this caveat a bit stronger.

"In terms of rainfall enhancement, **we are assuming** that the water quantities produced by the model microphysics are plausible. If they are representative, then the implications for these amounts are considerable."

We trust this makes the case more clearly. We also discussed the need to evaluate rainfall quantities in the future in the Summary and Outlook:

*"Confidence in rainfall enhancement should be further tested in further studies which assess simulation sensitivity, regional climate variability, and statistical analyses. Ensemble UAE simulations with varying model*

*physics including microphysics (e.g., Schwitalla et al., 2020; Fonseca et al., 2020, Taraphdar et al., 2021) would be particularly useful. Here, data assimilation, quantitative precipitation estimation, and rain radar analyses may also be employed (Bauer et al., 2015; Kawabata et al., 2018; Branch et al., 2020; Schwitalla & Wulfmeyer, 2014)."*

However, we will modify this a little to make it clear that work still needs to be done, including precipitation quantification evaluation in subsequent research:

*"Confidence in rainfall enhancement **and rainfall quantification** should be tested in further studies which assess simulation sensitivity, regional climate variability, and statistical analyses. Ensemble UAE simulations with varying model physics including microphysics (e.g., Schwitalla et al., 2020; Fonseca et al., 2020, Taraphdar et al., 2021) would be particularly useful. Here, data assimilation, quantitative precipitation estimation, and rain radar and satellite analyses may be employed (Bauer et al., 2015; Kawabata et al., 2018; Branch et al., 2020; Schwitalla & Wulfmeyer, 2014). The latter analyses, probably based on seasonal simulations, will be especially important to test further the plausibility of convective rainfall amounts, from a statistical point of view. For now, we have at least some confidence that WRF V4 has reproduced convective cells in a satisfactory way in the region (Fonseca et al. (2022)."*

Aside from the absolute representativeness of precipitation amounts though, the authors also consider that the relative rainfall enhancements between the different scenarios, and the clear intensification of convective processes, even from 20km scales upward, already represents a positive indicator for enhancement potential. In respect to convective processes we consider that the likely increases in updrafts, cloud development and convection initiation have been demonstrated convincingly here. In that respect, the convection evaluation paper of Fonseca et al. (2022) provides confidence in WRF's ability to produce convective cells, and in the right locations.

**Concluding remarks from the authors:**

Many thanks again for taking so much time to review our work and for your well-thought out and constructive comments. We feel that your comments have greatly improved our manuscript.

**New references**

Fonseca, R., Francis, D., Nelli, N., Farrah, S., Wehbe, Y., Al Hosari, T., & Al Mazroui, A. (2022). Assessment of the WRF Model as a Guidance Tool Into Cloud Seeding Operations in the United Arab Emirates. *Earth and Space Science*, *9*(5). https://doi.org/10.1029/2022EA002269

Temimi, M., Fonseca, R., Nelli, N., Weston, M., Thota, M., Valappil, V., Branch, O., Wizemann, H. D., Kondapalli, N. K., Wehbe, Y., Hosary, T. Al, Shalaby, A., Shamsi, N. Al, & Naqbi, H. Al. (2020). Assessing the impact of changes in land surface conditions on WRF predictions in arid regions. *Journal of Hydrometeorology*, *21*(12), 2829–2853. https://doi.org/10.1175/JHM-D-20-0083.1